# Cascading Contextual Assortment Bandits

**Hyun-jun Choi**
Seoul National University
Seoul, South Korea
nschj1@snu.ac.kr

**Rajan Udwani**
UC Berkeley
Berkeley, CA, USA
rudwani@berkeley.edu

**Min-hwan Oh**
Seoul National University
Seoul, South Korea
minoh@snu.ac.kr

## Abstract

We present a new combinatorial bandit model, the *cascading contextual assortment bandit*. This model serves as a generalization of both existing cascading bandits and assortment bandits, broadening their applicability in practice. For this model, we propose our first UCB bandit algorithm, `UCB-CCA`. We prove that this algorithm achieves a $T$-step regret upper-bound of $\tilde{\mathcal{O}}(\frac{1}{\kappa}d\sqrt{T})$, sharper than existing bounds for cascading contextual bandits by eliminating dependence on cascade length $K$. To improve the dependence on problem-dependent constant $\kappa$, we introduce our second algorithm, `UCB-CCA+`, which leverages a new Bernstein-type concentration result. This algorithm achieves $\tilde{\mathcal{O}}(d\sqrt{T})$ without dependence on $\kappa$ in the leading term. We substantiate our theoretical claims with numerical experiments, demonstrating the practical efficacy of our proposed methods.

## 1 Introduction

Sequential interactions between users and a recommender agent are often modeled as the multi-armed bandit problem or one of its variants [15]. In practice, a user typically encounters multiple items per round of interaction rather than a solitary item. Two popular models that capture this aspect are the cascading bandit [13; 14; 18] and the assortment bandit [4; 5; 7; 22], also often known as multinomial logistic bandits.

In the cascading bandit problem [13; 14; 18; 29; 25; 28], the agent selects a cascade of $K$ items from a total of $N$ items each round. These selected items are sequentially presented one at a time to a user. If the user clicks on a presented item, the cascading round ends. If not, the agent proceeds to reveal the next item from the cascade. This process continues until either the user clicks on an item or all $K$ items in the cascade have been presented without a click. Once a round ends, a next round commences with a newly selected list of $K$ items.

In the assortment bandit problem [4; 5; 7; 8; 9; 7; 22; 23], the agent presents an assortment of $M$ items all at once, then receives user choice feedback on the assortment. A user may opt for one of the $M$ items presented or choose none at all, concluding the round in either case. Both cascading and assortment bandit problems are significant combinatorial variations of the multi-armed bandit problem and have been extensively examined both theoretically and in practice.

However, a more commonly encountered scheme in real-world applications is a generalization of these two settings, where a *cascade of assortments* is sequentially revealed in each round. This approach is evident in video streaming services, where assortments of recommended contents are revealed as users scroll through webpages or mobile applications. Similar experiences can also be found in various online retail services and search engines. To address this, we propose a new interactive model, which we term *cascading assortment bandits*. In the cascading assortment bandits, the agent chooses a cascade of assortments that consists of $K$ assortments with each assortment containing $M$ items. The agent reveals one assortment at a time in the cascade. The cascade concludes if the user clicks on one of the items contained in a given assortment. If not, the agent proceeds

37th Conference on Neural Information Processing Systems (NeurIPS 2023).

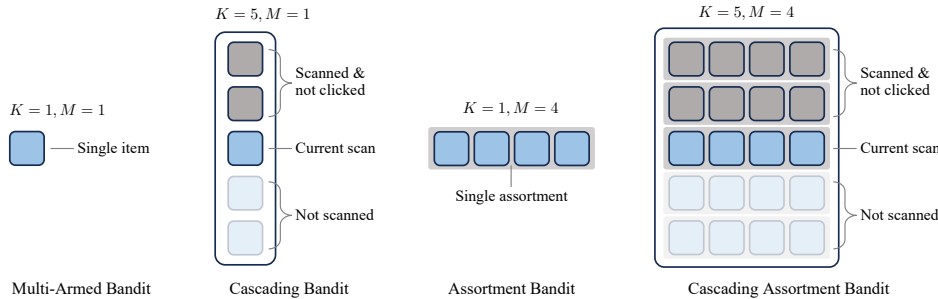

Figure 1: Comparisons between the cascading assortment bandit and the other combinatorial bandits. The cascading assortment bandit subsumes the multi-armed bandit ($K = 1, M = 1$), the cascading bandit ($K > 1, M = 1$), and the assortment bandit ($K = 1, M > 1$).

to unveil the subsequent assortment in the cascade. This cycle continues until either an assortment receives a click from the user or the agent depletes the pre-selected assortments.

The cascading assortment bandit problem is the strict generalization of both the cascading bandits model and the assortment bandits model. It is also a generalization of the simple multi-armed bandit problem. That is, if $K = 1$ and $M = 1$, the problem is the simple multi-armed bandit. If $K > 1$ and $M = 1$, the problem corresponds to the cascading bandit. If $K = 1$ and $M > 1$, we recover the assortment bandit problem. The illustrations on comparisons between the cascading assortment bandit and the other combinatorial bandits are presented in Figure 1.

In order to accommodate the generalization of the interactive model across items and assortments, we also incorporate the feature information of items and parametrization of a click model in the cascading assortment bandit model. Hence, we name the model as **cascading contextual assortment bandit** (see Section 2.2 for the formal definition of the problem setting).[1] Under this newly proposed combinatorial bandit model, we posit the following question:

*Can we design a provably efficient algorithm for cascading contextual assortment bandits?*

To address the question at hand, we first have to overcome the technical challenges inherent in each special case of our problem setting: the cascading contextual bandit and the contextual assortment bandit. Firstly, in the cascading contextual bandit [18; 25], a longstanding issue has been the suboptimal dependence on the cascade length, $K$. Intuitively, one would expect that as $K$ increases in the cascading model, the regret should either diminish or at least remain constant; performance deterioration should not occur. However, all existing regret bounds for cascading contextual bandits scale proportionally to $K$ [18; 29; 17; 28; 25]. This finding is not just counter-intuitive, but also suboptimal (for further discussions, refer to Section 2.4). As a result, (i) **eradicating the suboptimal dependence on cascade length** $K$ has been recognized as an open problem, even within the cascading contextual bandit setting.[2]

Further, in the context of assortment bandits, there is a widely recognized suboptimal dependence on the problem-specific constant $\kappa$, as demonstrated in the existing assortment bandit literature [9; 22; 23]. This problem-specific constant $\kappa$ (in Assumption 4.2) represents the curvature of the multinomial logit (MNL) function. Recent studies [24; 3] have demonstrated an improved dependence on $\kappa$, albeit only multiplied by logarithmic factors. However, this improvement comes at the expense of an increased dependence on the assortment size $M$, a conclusion that is both counter-intuitive and suboptimal. Thus, (ii) **decreasing the $\kappa$ dependence without escalating the dependence on** $M$ still poses an unresolved issue. While addressing either of the two challenges (i) and (ii) can be daunting individually, tackling both issues simultaneously poses an even greater challenge in both our algorithm design and regret analysis.

---

[1]Note that a non-contextual version of the cascading assortment bandit is a special case of the cascading contextual assortment bandit with a one-hot encoded feature vector for each item. Hence, when we aim to provide efficient algorithms for cascading contextual assortment bandit, we also address the non-contextual cascading assortment bandit which has not been studied previously.

[2]A concurrent work [20] addresses this suboptimal dependence on $K$ under the linear model assumption. Our work tackles this challenge under the MNL model in a more general setting.

Table 1: Comparisons of algorithms for contextual cascade and assortment bandits as well as for cascading contextual assortment bandits. $N$ is the number of ground items, $K$ is a length of cascade, $d$ is a dimension of feature vectors and $T$ is total rounds. $\kappa$ is a problem-dependent parameter for the MNL model. See Appendix A for more discussions.

| Algorithm | Context | Cascade | Assortment | Click Model | Regret Bound |
|---|---|---|---|---|---|
| CombCascade [14] | $\times$ | $\bigcirc$ | $\times$ | $\times$ | $\tilde{\mathcal{O}}(\sqrt{KNT})$ |
| C³-UCB [18] | $\bigcirc$ | $\bigcirc$ | $\times$ | Linear | $\tilde{\mathcal{O}}(d\sqrt{KT})$ |
| EE-MNL [5] | $\times$ | $\times$ | $\bigcirc$ | MNL | $\tilde{\mathcal{O}}(\sqrt{NT})$ |
| TS-MNL [22] | $\bigcirc$ | $\times$ | $\bigcirc$ | MNL | $\tilde{\mathcal{O}}(\frac{1}{\kappa}d^{3/2}\sqrt{T})$ |
| UCB-MNL [23] | $\bigcirc$ | $\times$ | $\bigcirc$ | MNL | $\tilde{\mathcal{O}}(\frac{1}{\kappa}d\sqrt{T})$ |
| LinTS-Cascade [28] | $\bigcirc$ | $\bigcirc$ | $\times$ | Linear | $\tilde{\mathcal{O}}(d^{3/2}K\sqrt{T})$ |
| CascadeWOFUL [25] | $\bigcirc$ | $\bigcirc$ | $\times$ | Linear | $\tilde{\mathcal{O}}(\sqrt{d^2T + dTK})$ |
| VAC²-UCB [20] | $\bigcirc$ | $\bigcirc$ | $\times$ | Linear | $\tilde{\mathcal{O}}(d\sqrt{T})$ |
| UCB-CCA (Algorithm 1) | $\bigcirc$ | $\bigcirc$ | $\bigcirc$ | MNL | $\tilde{\mathcal{O}}(\frac{1}{\kappa}d\sqrt{T})$ |
| UCB-CCA+ (Algorithm 2) | $\bigcirc$ | $\bigcirc$ | $\bigcirc$ | MNL | $\tilde{\mathcal{O}}(d\sqrt{T})$ |

To this end, we design novel upper confidence bound (UCB) algorithms for contextual cascading assortment bandits, tackling both technical challenges. We show that our proposed algorithms achieve provable guarantees on regret performances overcoming the longstanding technical challenges. Our regret bounds show sharper results than those of the existing contextual cascading bandits or assortment bandits. We corroborate our theoretical claims through numerical experiments, thus ensuring that both our newly proposed bandit framework and the proposed algorithms establish provable efficiency and practical applicability.

Our main contributions are summarized as follows.

- We formulate a general combinatorial bandit model, named *cascading contextual assortment bandit* that encompasses the existing cascading bandits and assortment bandits. This novel problem setting is observed in many practical applications.

- We first propose a UCB bandit algorithm UCB-CCA for the cascading contextual assortment bandit and establish the $T$-step regret upper-bound of $\tilde{\mathcal{O}}(\frac{1}{\kappa}d\sqrt{T})$ (in Theorem 4.3). This regret bound is tighter than the existing bounds for cascading contextual bandits, where we not only remove the longstanding, unnecessary dependence on $K$ but also establish the result without dependence on $M$.

- While UCB-CCA is an efficient algorithm achieving both the statistical efficiency and practical performances (shown in Section 7), its regret bound includes dependence on the inverse of a problem-dependent constant $\kappa$, which can be potentially large in the worst case. To improve the dependence on $\kappa$, we propose our second algorithm UCB-CCA+, which exploits a new Bernstein-type concentration result, taking into account the effects of the local curvature of the MNL model. We prove that UCB-CCA+ achieves $\tilde{\mathcal{O}}(d\sqrt{T})$ without the dependence on $\kappa$ in the leading term (only scaling with logarithmic factors), hence significantly improving the regret of UCB-CCA without increasing the other dependencies. Hence, we successfully solve the two technical challenges (i) and (ii) mentioned above.

- As an independent contribution, we prove that a greedy algorithm for the cascading assortment optimization problem gives a 0.5 approximation of the optimal solution (discussed in Section 6). To our best knowledge, this is the first rigorous result showing the approximation guarantee even for the contextual cascading bandit problem, instead of simply assuming access to an approximation optimization oracle.

- We evaluate our proposed methods in numerical experiments and show that the practical performances support our theoretical claims. Hence, our proposed algorithms along with our newly proposed bandit model establish provable efficiency and practical applicability.

## 2 Preliminaries

### 2.1 Notation

Define $[n]$ as a set of positive integers from $1$ to $n$. Let $|\cdot|$ be the length of a sequence or the cardinality of a set. For a vector $x \in \mathbb{R}^d$, we denote the $\ell_2$-norm of $x$ as $||x||_2$ and the $V$-weighted norm of $x$ for a positive-definite matrix $V$ as $||x||_V = \sqrt{x^\top V x}$. The determinant and trace of a matrix $V$ are $\det(V)$ and $\text{trace}(V)$, respectively. $\lambda_{\min}(V)$ denotes the minimum eigenvalue of a matrix $V$.

### 2.2 Cascading Contextual Assortment Bandit Problem

Consider $[N]$, a set of $N$ items. Let $\mathcal{A}$ be a set of candidate assortments of items with size $M$, i.e., $\mathcal{A} := \{A \subseteq [N] : |A| = M\}$. A cascade $S$ is an ordered sequence of $K$ assortments chosen from $\mathcal{A}$ where all the items in these $K$ assortments are distinct. Then, the set of all feasible cascades $\mathcal{S}$ can be defined as follows.

$$\mathcal{S} := \left\{ S = (A_1, ..., A_K) \mid A_k \in \mathcal{A} \text{ for all } k \in [K], \cap_{k=1}^K A_k = \emptyset \right\}$$

At round $t$, feature vectors $\{x_{ti} \in \mathbb{R}^d, i \in [N]\}$ for every item are revealed to the decision-making agent. Each feature vector $x_{ti}$ may contain the contextual information of the user at round $t$ and the item $i$. After observing this contextual information, at round $t$, the agent recommends a cascade $S_t = (A_{tk})_{k \in [K]}$ to the user, where $A_{tk} \in \mathcal{A}$ represents the $k$-th assortment of the cascade at round $t$. The user scans the assortments in $S_t$ one by one. If the items in $A_{tk}$ do not attract the user, the user moves on to the next assortment $A_{t,k+1}$. The user stops at the $O_t$-th assortment when the user is attracted by an item in the $O_t$-th assortment and clicks on the item.

After the user clicks on the item, the agent observes a sequence of user choices $y_t = (y_{tk})_{k \in [O_t]}$ where a binary vector $y_{tk} = (y_{tk0}, y_{tk1}, ..., y_{tkM})$ represents user choices on assortment $A_{tk}$. Let $y_{tkm} = 1$ if the $m$-th item $i_m$ in $A_{tk}$ is clicked by the user, and $y_{tkm} = 0$ for items that are not clicked on. For each assortment, there is an *outside option*. That is, there is a probability that the user may not click any of the items in $A_{tk}$. If the user does not choose any items, $y_{tk0} = 1$ and $y_{tkm} = 0$ for all $m \in [M]$. The user choice for each assortment is given by the multinomial logit (MNL) choice model [21]. For this MNL model, there is an *unknown* time-invariant parameter $\theta^* \in \mathbb{R}^d$. We define the true weight of item $i$ in round $t$ as $w_{ti}^* := x_{ti}^\top \theta^*$. Also, we let the vector representation of the weights be defined as $w_t^* := (w_{ti}^*)_{i \in [N]}$ for convenience.

Under this model, the user's click probability of the $m$-th item in $A_{tk}$ and the probability of the outside option in $A_{tk}$ is given respectively by

$$p_t(i_m | A_{tk}, w_t^*) = \frac{\exp(w_{ti_m}^*)}{1 + \sum_{j \in A_{tk}} \exp(w_{tj}^*)} \quad \text{and} \quad p_t(i_0 | A_{tk}, w_t^*) = \frac{1}{1 + \sum_{j \in A_{tk}} \exp(w_{tj}^*)}$$

where item $i_0$ represents the outside option. The user choice $y_{tk}$ is sampled from the multinomial distribution, $y_{tk} \sim \text{MNL}\{1, (p_t(i_m | A_{tk}, w_t^*))_{m=0}^M\}$, where the argument $1$ indicates that $y_{tk}$ is a single-trial sample. Hence, $\sum_{m=1}^M y_{tkm}$ is always $1$. Also, we denote measurement noise as $\epsilon_{tkm} := y_{tkm} - p_t(i_m | A_{tk}, w_t^*)$. Since $\epsilon_{tkm}$ is bounded in $[0, 1]$, $\epsilon_{tkm}$ is $\sigma^2$-sub-Gaussian with $\sigma^2 = 1/4$. It is important to note that $\epsilon_{tkm}$ across items in the same assortment is not independent due to the substitution effect in the MNL model.

The expected reward function of a combinatorial action $S_t$ based on $w_t^*$ is given by

$$f(S_t, w_t^*) = \sum_{k=1}^K \left\{ \prod_{\dot{k}=1}^{k-1} p_t(i_0 | A_{t\dot{k}}, w_t^*) \right\} \sum_{i \in A_{tk}} p_t(i | A_{tk}, w_t^*) = 1 - \prod_{k=1}^{|S_t|} p_t(i_0 | A_{tk}, w_t^*).$$

The formulation above is also known as the cascade model with disjunctive objective, where the user stops at the *first attractive* item [13; 14; 18].

### 2.3 $\alpha$-Approximation Oracle and $\alpha$-Regret

The exact combinatorial optimization to compute an optimal cascade of assortments can be computationally expensive. Therefore, we allow for approximate optimization. We assume that the

agent has access to an $\alpha$-optimization oracle to compute a $\alpha$-approximation solution of the cascade optimization problem with $\alpha \leq 1$. For approximate optimization, we prove that a greedy selection for the cascading assortment optimization problem gives a 0.5 approximation of the optimal solution, which may be of independent interest (see Section 6).

Formally, for a given an $\alpha$-optimization oracle and a weight parameter $w$, the oracle outputs an approximately optimal cascade $\hat{S}^* = \mathbb{O}^\alpha(w) \in \mathcal{S}$ satisfying $f(\hat{S}^*, w) \geq \alpha f(S^*, w)$ where $S^* \in \arg\max_{S \in \mathcal{S}} f(S, w)$ is an optimal assortment without approximation. The instantaneous $\alpha$-regret of cascade $S_t$ in round $t$ is defined as $\mathcal{R}^\alpha(t, S_t) := \mathbb{E}[\alpha f(S_t^*, w_t^*) - f(S_t, w_t^*)]$ where $S_t^* \in \arg\max_{S \in \mathcal{S}} f(S, w_t^*)$ is an true optimal assortment. Then, the goal of the agent is to minimize the cumulative $\alpha$-regret defined as

$$\mathcal{R}^\alpha(T) := \sum_{t=1}^T \mathcal{R}^\alpha(t, S_t) = \sum_{t=1}^T \mathbb{E}[\alpha f(S_t^*, w_t^*) - f(S_t, w_t^*)].$$

### 2.4 Suboptimal Dependence on Problem Dependent Parameters

In this subsection, we discuss the main technical challenges faced in the regret analysis of our problem setting. In particular, the suboptimal dependence on cascade length $K$ has been a long-standing open problem even in contextual cascading bandits.

#### 2.4.1 Dependence on Length of Cascade $K$

The previous literature on the contextual cascading bandits [18; 17; 26; 25] bounds the instantaneous regret in each round, utilizing the monotonicity and Lipschitz continuity of the expected reward function $f$. A simple adaptation of the previously known techniques to our problem would result in the following upper bound for the instantaneous regret $\mathcal{R}^\alpha(t, S_t)$ for $S_t = (A_{t1}, A_{t2}, ..., A_{tK})$.

$$\mathcal{R}^\alpha(t, S_t) \leq \sum_{k=1}^K \sum_{i \in A_{tk}} \beta_t ||x_{ti}||_{V_{t-1}^{-1}} \tag{1}$$

where $\beta_t$ is a suitable confidence radius chosen by an algorithm, and $V_t$ is a positive definite gram matrix. Then, the dependence on the length of cascade $K$ and the assortment size $M$ would appear in the regret bound after summing the right-hand side of Eq.(1) over the time horizon and applying the Cauchy-Schwarz inequality. Because of this reason, even when $M = 1$, there still exists dependence on $K$ which appears in the regret bounds of all previous contextual cascading bandits (see Table 1).

To overcome this challenge, we present a new Lipschitz continuity of the expected reward function to derive the regret bound independent of $M$ and $K$ by replacing the summation with the maximum over assortments and a cascade (see Section 4.3 for more details).

#### 2.4.2 Dependence on Worst-Case Scanning Probability

Analogous to the existing algorithms for the contextual cascading bandits [18; 26], the gram matrix $V_t$ contains the rank-1 matrices of observed items accumulated up to round $t$, i.e., $V_t = \sum_{\tau=1}^t \sum_{k=1}^{O_\tau} \sum_{i \in A_{\tau k}} x_{\tau i} x_{\tau i}^\top + \lambda I$. However, there exists an out-of-control issue, that is, the summation of the rank-1 matrices over $O_t + 1$ to $|S_t|$ in Eq.(1) is not included in the gram matrix $V_t$. Note that this issue also arises in cascading contextual assortment bandits. Adapting a technique used in the existing literature [18] to mitigates this issue, let $p_{t,S_t}$ be the probability of examining all assortments in $S_t$ and $p^* = \min_{t \in [T]} \min_{S \in \mathcal{S}} p_{t,S_t}$ be the worst-case probability of examining a cascade over all rounds and all feasible cascades. A simple adaptation of the existing methods would result in the following bound for the expected instantaneous regret.

$$\mathbb{E}[\mathcal{R}^\alpha(t, S_t)] = \mathbb{E}\left[\mathcal{R}^\alpha(t, S_t)\mathbb{E}\left[\frac{1}{p_{t,S_t}}\mathbb{1}\{O_t = |S_t|\} \mid S_t\right]\right] \leq \frac{1}{p^*}\mathbb{E}[\mathcal{R}^\alpha(t, S_t)\mathbb{1}\{O_t = |S_t|\}].$$

This concedes the dependence on $p^*$, which can be exponentially small in the worst case. We overcome this challenge by designing an algorithm that offers the assortment containing the most uncertain item as the first assortment in a cascade. We discuss this salient feature of the proposed algorithm in more detail in Section 3.2.

---

**Algorithm 1** UCB-CCA

---

**Input**: confidence radius $\beta_t$ and ridge penalty parameter $\lambda \geq 1$

1: **for** $t = 1, \ldots, T$ **do**
2:     Observe $x_{ti}$ for all $i \in [N]$
3:     Compute $u_{ti} = x_{ti}^\top \hat{\theta}_{t-1} + \beta_{t-1} ||x_{ti}||_{V_{t-1}^{-1}}$ for all $i \in [N]$
4:     Compute a candidate cascade $S'_t \leftarrow (A'_{tk})_{k \in [K]} = \mathbb{O}^\alpha(u_t)$
5:     Find optimistic exposure assortment index $k^*$ in $(k^*, i^*) = \underset{k \in [K], i \in A'_{tk}}{\operatorname{argmax}} ||x_{ti}||_{V_{t-1}^{-1}}$
6:     Optimistic exposure swap $S_t \leftarrow (A_{tk})_{k \in [K]}$ where $A_{tk} := \begin{cases} A'_{tk^*} & \text{if} \quad k = 1 \\ A'_{t1} & \text{if} \quad k = k^* \\ A'_{tk} & \text{otherwise} \end{cases}$
7:     Offer $S_t$, and observe user feedback $O_t$ and $y_t = (y_{tk})_{k \in [O_t]}$
8:     Update $V_t \leftarrow V_{t-1} + \sum_{k=1}^{O_t} \sum_{i \in A_{tk}} x_{ti} x_{ti}^\top$
9:     Compute the regularized MLE $\hat{\theta}_t$ by solving $\nabla_\theta \left[\ell_t(\theta) + \frac{\lambda}{2} ||\theta||_2^2\right] = \mathbf{0}$
10: **end for**

---

## 3 Algorithm: UCB-CCA

### 3.1 Upper Confidence Bounds and Confidence Set

UCB-CCA utilizes the upper confidence bounds (UCB) technique [6; 1; 16] to compute an optimistic action based on optimistic estimates of each item's weight, $u_{ti} = x_{ti}^\top \hat{\theta}_{t-1} + \beta_{t-1}(\delta) ||x_{ti}||_{V_{t-1}^{-1}}$ for all $i \in [N]$. The confidence radius $\beta_t(\delta)$ is specified to maintain a high-probability confidence set $C_t(\delta)$ for the unknown parameter $\theta^*$, although the algorithm does not explicitly compute $C_t(\delta)$.

$$C_t(\delta) := \left\{\theta \in \mathbb{R}^d : ||\hat{\theta}_t - \theta||_{V_t} \leq \beta_t(\delta)\right\}.$$

Setting a proper confidence radius $\beta_t(\delta)$ can guarantee that $\theta^*$ lies within the confidence set with probability $1 - \delta$. On the event that $\theta^* \in C_t(\delta)$, the UCB weight $u_{ti}$ serves as an upper bound of a true weight $w_{ti}^* := x_{ti}^\top \theta^*$ for every item $i \in [N]$. We denote the UCB weight vector as $u_t = (u_{ti})_{i \in [N]}$ for convenience.

### 3.2 Optimistic Exposure Swapping

A distinctive element of UCB-CCA is what we call *optimistic exposure swapping*, a procedure crucial for eliminating dependence on the worst-case scanning probability, as elaborated in Section 2.4.2. This technique strategically positions the assortment containing the item with the highest uncertainty among the top $MK$ items in the first slot of the cascade of assortments.

In each round $t$, the $\alpha$-approximate oracle $\mathbb{O}^\alpha(u_t)$ outputs a *candidate* cascade $S'_t$, determined by the UCB weights $u_t$. It is important to note that $S'_t$ is not immediately presented to the user. Instead, after $S'_t$ is derived using the optimization oracle $\mathbb{O}^\alpha(u_t)$, the algorithm identifies the index $k^*$ of an assortment that includes the item with the largest $||x_{ti}||_{V_{t-1}^{-1}}$ in $S'_t$.

Subsequently, the algorithm swaps the positions: the assortment $A'_{tk^*}$ is moved to the top of $S_t$, becoming $A_{t1}$, and the initially top assortment $A'_{t1}$ in $S'_t$ is relocated to the $k^*$-th position of $S_t$, now $A_{tk^*}$. The positions of the other assortments remain the same, that is, $A_{tk} = A'_{tk}$ for all $k \in [K] \setminus \{1, k^*\}$.[3] This procedure is viable as the expected reward is unaffected by the display order of assortments in the cascade, as shown in Lemma 4.5.

### 3.3 Regularized Maximum Likelihood Estimation

UCB-CCA computes a regularized maximum likelihood estimate of the unknown parameter $\theta^*$. The negative log-likelihood is given by $\ell_t(\theta) = -\sum_{\tau=1}^{t-1} \sum_{k=1}^{O_\tau} \sum_{m=0}^{M} y_{\tau km} \log p_\tau(i_m | A_{\tau k}, w_\tau)$, where

---

[3]While the swapping occurs between the first and the $k^*$-th positions for specificity, it is sufficient to place $A'_{tk^*}$ at the top position of $S_t$. The sequence of the remaining assortments is not critical.

$w_t = (w_{ti})_{i \in [N]}$ is a weight vector, and its element is $w_{ti} = x_{ti}^\top \theta$. For penalty parameter $\lambda \geq 1$, the $\ell_2$-regularized MLE is given by

$$\hat{\theta}_t = \operatorname*{argmin}_\theta \left[ \ell_t(\theta) + \frac{\lambda}{2} ||\theta||_2^2 \right] = \operatorname*{argmin}_\theta \left[ -\sum_{\tau=1}^{t-1} \sum_{k=1}^{O_\tau} \sum_{m=0}^{M} y_{\tau km} \log p_\tau(i_m | A_{\tau k}, w_\tau) + \frac{\lambda}{2} ||\theta||_2^2 \right]. \tag{2}$$

## 4 Regret Analysis of `UCB-CCA`

### 4.1 Regularity Condition

**Assumption 4.1.** $||x||_{ti} \leq 1$ for all round $t$ and items $i \in [N]$, and also $||\theta^*|| \leq 1$.

**Assumption 4.2.** There exists $\kappa > 0$ such that for all $t \in [T]$, any assortment $A \in \mathcal{A}$, and any item $i \in A$, $\inf_{\theta \in \mathbb{R}^d} p_t(i|A, w) p_t(i_0|A, w) \geq \kappa$, where $w = (w_i)_{i \in [N]}$ and $w_i = x_i^\top \theta$.

**Discussion of Assumptions.** Assumption 4.1 makes the regret bound independent on the scale of the feature vector and parameter. This is the standard assumption used in the contextual bandit literature [1; 18; 22]. Assumption 4.2 is the standard regularity assumption in the contextual assortment bandit literature [8; 27; 22; 7; 23], adapted from the standard assumption for the link function in the generalized linear contextual bandit literature [16] to ensure that the Fisher information matrix is non-singular.

### 4.2 Regret Bound of `UCB-CCA`

**Theorem 4.3** ($\alpha$-regret upper bound of `UCB-CCA`)**.** *Suppose Assumptions 4.1 and 4.2 hold, and we run `UCB-CCA` for total $T$ rounds with $\beta_t = \frac{1}{2\kappa} \sqrt{d \log \left( 1 + \frac{tKM}{d\lambda} \right) + 4 \log t} + \frac{\sqrt{\lambda}}{\kappa}$ and with $\lambda \geq 1$, Then, the $\alpha$-regret of `UCB-CCA` is upper-bounded by*

$$\mathcal{R}^\alpha(T) \leq \left( \frac{K}{K+1} \right)^{K+1} \left[ \frac{1}{2\kappa} \sqrt{d \log \left( 1 + \frac{TKM}{d\lambda} \right) + 4 \log T} + \frac{\sqrt{\lambda}}{\kappa} \right] \sqrt{2dT \ln \left( 1 + \frac{TKM}{\lambda d} \right)}.$$

**Discussion of Theorem 4.3.** Theorem 4.3 establishes that `UCB-CCA` achieves a regret bound of $\tilde{\mathcal{O}}\left( \frac{d}{\kappa} \sqrt{T} \right)$. Notably, this regret bound removes dependence on $p^*$ completely and removes polynomial dependence on $K$, achieving the best-known bound in contextual cascading bandits [18; 25; 20], a special case of our problem setting. Apart from the generalization we consider in this work, a key distinction between our work and the previous contextual cascading bandit models lies in our adoption of the MNL model, as opposed to the linear model assumed by the existing literature. This model choice introduces a dependence on the parameter $\kappa$ within the regret bound of `UCB-CCA`, which is an aspect we address and refine in subsequent sections. It is essential to highlight that our work tackles a more general problem yet achieves improved bounds concerning the key problem parameters previously considered suboptimal. The factor comprised of the length of the cascade, $\left( K/(K+1) \right)^{K+1}$, in Theorem 4.3 is also notable, which is bounded above by 1 regardless of the value of $K$. Consequently, the regret bound does not increase polynomially with $K$, ensuring scalability.

### 4.3 Proof Outline

In this subsection, we present the proof sketch of Theorem 4.3. One of the key components of the regret analysis that enables carving off the dependence on $K$ is the following lemma.

**Lemma 4.4** (Maximal Lipschitz continuity)**.** *Suppose $u_{ti} \geq w_{ti}^*$ for all $i \in [N]$. Then*

$$f(S_t, u_t) - f(S_t, w_t^*) \leq \left( \frac{K}{K+1} \right)^{K+1} \max_{A_{tk} \in S_t} \max_{i \in A_{tk}} (u_{ti} - w_{ti}^*). \tag{3}$$

Lemma 4.4 demonstrates that the difference between the reward functions under the UCB parameter and the true parameter is upper bounded by the maximal difference between the UCB and true parameters. This implies that the regret bound remains unaffected by increases in the cascade length $K$ or the assortment size $M$.

Eliminating the dependence on $p^*$ is another key element of our analysis. To this end, we first show that the order of assortments in the cascade model with the disjunctive objective does not affect the expected reward. We formalize this property in the following lemma.

**Lemma 4.5.** *Let $p_k$ be the probability that the user clicks on any item in $A_k$. Given a collection of assortments $\{A_1, \cdots, A_K\}$ with probabilities $\{p_1, \cdots, p_K\}$, their order of display does not matter. Further, for every permutation $\rho : [K] \to [K]$, we have*

$$\sum_{k \in [K]} p_k \prod_{\dot{k} < k} (1 - p_{\dot{k}}) = 1 - \prod_{k \in [K]} (1 - p_k) = \sum_{k \in [K]} p_{\rho^{-1}(k)} \prod_{\dot{k} < k} \left(1 - p_{\rho^{-1}(\dot{k})}\right).$$

Consolidating these key results, we proceed to bound the cumulative regret. We begin by leveraging the monotonicity of the expected reward function and the definition of the $\alpha$-approximate optimization oracle to bound the cumulative regret.

$$\mathcal{R}^\alpha(T) \leq \mathbb{E}\left[\sum_{t=1}^{T} f(S_t, u_t) - f(S_t, w_t^*)\right] \leq C_K \mathbb{E}\left[\sum_{t=1}^{T} \max_{A_{tk} \in S_t} \max_{i \in A_{tk}} (u_{ti} - w_{ti}^*)\right]$$

$$\leq 2C_K \mathbb{E}\left[\beta_T \sum_{t=1}^{T} \max_{A_{tk} \in S_t} \max_{i \in A_{tk}} ||x_{ti}||_{V_{t-1}^{-1}}\right] = 2C_K \mathbb{E}\left[\beta_T \sum_{t=1}^{T} \max_{k \in [O_t]} \max_{i \in A_{tk}} ||x_{ti}||_{V_{t-1}^{-1}}\right]. \quad (4)$$

The second inequality is from Lemma 4.4, letting $C_K := (K/(K+1))^{K+1}$. The third inequality is given by the concentration of the UCB weights (see Lemma B.5). Note that the assortment including the item with the largest value of $||x_{ti}||_{V_t^{-1}}$ is always examined by the user since it is included in the first assortment of $S_t$ by the optimistic exposure swapping technique as described in Section 3.2. Note that a change in the order of assortments incurred by the optimistic exposure swapping does not affect the expected reward which is shown in Lemma 4.5. Hence, for every round $t \in [T]$, we obtain

$$\max_{A_{tk} \in S_t} \max_{i \in A_{tk}} ||x_{ti}||_{V_{t-1}^{-1}} = \max_{k \in [O_t]} \max_{i \in A_{tk}} ||x_{ti}||_{V_{t-1}^{-1}}. \quad (5)$$

Therefore, the last equality in Eq.(4) is given by Eq.(5). Then, we can apply the maximal version of elliptical potential lemma (see Lemma B.8) to bound the cumulative regret.

# 5 Improved Dependence on $\kappa$

While `UCB-CCA` achieves the regret bound of $\tilde{\mathcal{O}}\left(\frac{1}{\kappa} d \sqrt{T}\right)$ improving dependence on $K$, the bound includes the problem-dependent constant $\kappa$. This implies a potential risk of the regret bound becoming large when $\kappa$ becomes very small. In order to circumvent this challenge, we propose a new *optimism in the face of uncertainty (OFU)* algorithm, `UCB-CCA+`, which exhibits a regret bound that is independent of $\kappa$ in the leading term. The pseudocode of `UCB-CCA+` is detailed in Algorithm 2.

## 5.1 Algorithm: `UCB-CCA+`

### 5.1.1 Confidence Set

`UCB-CCA+` computes a regularized MLE $\hat{\theta}_t$, following the same procedure described in Section 3.3. Then, the algorithm constructs a new confidence set centered around $\hat{\theta}_t$ utilizing a Bernstein-type tail inequality for self-normalized martingales [11; 3]. Nevertheless, a simple adaptation of the previous approaches may incur increased dependence on $M$. Hence, a more intricate analysis and refined algorithmic strategy are imperative to effectively address this challenge.

First, we define $g_t(\theta) := \sum_{\tau=1}^{t} \sum_{k \in [O_\tau]} \sum_{i \in A_{\tau k}} p_\tau(i|A_{\tau k}, w_\tau) x_{\tau i} + \lambda_t \theta$ where $w_t = (w_{ti})_{i \in [N]}$ and $w_{ti} = x_{ti}^\top \theta$. We also denote the partial derivative of $p_t(i|A_{tk}, w_t)$ with respect to $w_{ti}$ as $\dot{p}_t(i|A_{\tau k}, w_\tau) := p_t(i|A_{tk}, w_t) p_t(i_0|A_{tk}, w_t)$. Additionally, we define the new design matrix containing local information, denoted as $H_t(\theta) := \sum_{\tau=1}^{t} \sum_{k \in [O_\tau]} \sum_{i \in A_{\tau k}} \dot{p}_\tau(i|A_{\tau k}, w_\tau) x_{\tau i} x_{\tau i}^\top + \lambda_t I_d$, and, for convenience, $H_t := \sum_{\tau=1}^{t} \sum_{k \in [O_\tau]} \sum_{i \in A_{\tau k}} \dot{p}_\tau(i|A_{\tau k}, w_\tau^*) x_{\tau i} x_{\tau i}^\top + \lambda_t I_d$. Then, the algorithm constructs a confidence set as below:

$$B_t(\delta) := \left\{\theta \in \mathbb{R}^d : ||g_t(\hat{\theta}_t) - g_t(\theta)||_{H_t^{-1}(\theta)} \leq \gamma_t(\delta)\right\} \quad (6)$$

---

**Algorithm 2** UCB-CCA+

---

**Input**: confidence radius $\gamma_t$ and ridge penalty parameter $\lambda \geq 1$

1: **for** $t = 1, \dots, T$ **do**
2:     Observe $x_{ti}$ for all $i \in [N]$
3:     Construct a confience set $B_{t-1}(\delta)$ as defined in Eq.(6)
4:     Compute a candidate cascade $(S_t' = (A_{tk}')_{k \in [K]}, \theta_t) = \arg\max_{S \in \mathcal{S}, \theta \in B_t(\delta)} f(S, w_t)$
5:     Find optimistic exposure assortments $L_{t,H}$ and $L_{t,V}$ (and their positions $h$ and $v$) in $S_t'$
6:     $S_t \leftarrow (A_{tk})_{k \in [K]}$ where $A_{tk} = \begin{cases} L_{t,H} & \text{if} \quad k = 1 \\ L_{t,V} & \text{if} \quad k = 2 \\ A_{t1}' & \text{if} \quad k = h \\ A_{t2}' & \text{if} \quad k = v \\ A_{tk}' & \text{otherwise} \end{cases}$
7:     Offer $S_t$ and observe $O_t, y_t = (y_{tk})_{k \in [O_t]}$
8:     Update $H_t \leftarrow H_{t-1} + \sum_{k=1}^{O_t} \sum_{i \in A_{tk}} \dot{p}_t(i|A_{tk}, w_t^*) x_{ti} x_{ti}^\top$
9:     Update $V_t \leftarrow V_{t-1} + \sum_{k=1}^{O_t} \sum_{i \in A_{tk}} x_{ti} x_{ti}^\top$
10:     Compute the regularized MLE $\hat{\theta}_t$ by solving $\nabla_\theta \left[ \ell_t(\theta) + \frac{\lambda}{2} ||\theta||_2^2 \right] = \mathbf{0}$
11: **end for**

---

where the confidence radius $\gamma_t(\delta)$ is suitably specified to ensure that the true parameter $\theta^*$ lies in the confidence set $B_t(\delta)$ with high probability. On the event of $\theta^* \in B_t(\delta)$, The following lemma bounds the weighted $\ell_2$-norm of the difference between $\theta$ and $\theta^*$.

**Lemma 5.1.** *Suppose $\theta^* \in B_t(\delta)$. Then, for any $\theta \in B_t(\delta)$, we have $||\theta - \theta^*||_{H_t} \leq 6\gamma_t(\delta)$.*

### 5.1.2 Doubly Optimistic Exposure Swapping

UCB-CCA+ still faces the challenge outlined in Section 2.4.2. The complication is exacerbated for UCB-CCA+ as the algorithm concurrently updates two gram matrices, $H_t$ and $V_t$, which only contain the information of the observed items. Building upon the technique of optimistic exposure swapping detailed in Section 3.2, in each round $t$, UCB-CCA+ assigns the assortment $L_{t,H}$ — containing the item with the largest uncertainty with respect to $H_{t-1}$ among the top $KM$ items — to the first slot of cascade $S_t$. Similarly, the algorithm places $L_{t,V}$ — with the item that has the largest uncertainty with respect to $V_{t-1}$ — in the second slot of $S_t$.

### 5.2 Regret Analysis of UCB-CCA+

**Theorem 5.2** (Regret upper bound of UCB-CCA+). *Suppose Assumptions 4.1 and 4.2 hold, and we run UCB-CCA+ for total $T$ rounds with $\lambda \geq 1$ and $\gamma_t(\delta) := \frac{3\sqrt{\lambda_t}}{2} + \frac{2}{\sqrt{\lambda_t}} \log\left( \frac{(\lambda_t + KMt/d)^{d/2} \lambda_t^{-d/2}}{\delta} \right) + \frac{2d}{\sqrt{\lambda_t}} \log 2$ with $\delta = \frac{1}{t^2}$. Then, the regret of UCB-CCA+ is upper-bounded by*

$$\mathcal{R}^\alpha(T) \leq C_1 \gamma_T(\delta) \sqrt{2dT \log\left(1 + \frac{KMT}{d\lambda}\right)} + \frac{C_2}{\kappa} \gamma_T(\delta)^2 d \log\left(1 + \frac{KMT}{d\lambda}\right)$$

*where $C_1 = 36$ and $C_2 = 216(1 + Me)$.*

**Discussion of Theorem 5.2.** Theorem 5.2 establishes the regret bound of $\tilde{\mathcal{O}}(d\sqrt{T})$. The leading term of the regret bound is independent of $\kappa$. Although the second term exhibits dependence on $\kappa$, the term only scales logarithmically in $T$, whose comparative effect diminishes as $t$ increases compared to the leading term. Hence, the worst-case regret guarantee of UCB-CCA+ improves from that of UCB-CCA. The comprehensive proof of Theorem 5.2 is provided in the appendix.

## 6 Approximation Algorithm for Optimal Combinatorial Action

In this section, we show that computing the optimal cascade is, in general, a weakly NP-hard problem and give a (fast) polynomial time algorithm that the agent can use to compute a $0.5$-approximation to

the optimal cascade for any weight $w$. We assume that the MNL weights are given and consider the problem of finding the cascade that maximizes the expected reward. This is an optimization problem on selecting a sequence of $K$ assortments with size $M$ each from a ground set of $N$ items. The number of feasible cascades is $O\left(\binom{N}{M}^K\right)$. In fact, we establish the following hardness result.

**Lemma 6.1.** *For general $M$, the optimization problem is weakly NP-hard even for $K = 2$.*

The hardness of our problem follows from the hardness result of [19] for a related setting of unconstrained cascade optimization where the size of each assortment can be arbitrary. In light of this hardness, we turn our attention to finding fast approximation algorithms for the problem. While there has been a lot of recent work on the unconstrained cascade optimization problem (see [19; 12; 10] and the references therein), none of the previous algorithms apply to our constrained setting (where the size of each assortment is $M$).

We consider the following greedy approach for the problem. Consider arbitrary weights at round $t$, i.e. $w_t = (w_{ti})_{i \in [N]}$, is given. We order the items in $[N]$ in decreasing order of given weights and consider the following cascade of assortments.

$$D_1 = \{1, 2, ..., M\}, D_2 = \{M + 1, M + 2, ..., 2M\}, \cdots, D_K = \{(K - 1)M + 1, \cdots, KM\}$$

We call these assortments "decreasing order assortments". Let OPT denote the value of the optimal solution to the problem. We establish that showing these assortments in any possible sequences gives a good approximation to the problem.

**Lemma 6.2.** *Let* OPT *denote the overall click probability in the optimal solution. The decreasing order assortments $D_1, ..., D_K$, shown in any order, have overall click probability at least $0.5$ OPT.*

We present the proof in Appendix E. We also show that our algorithm is optimal for $M = 1$, which captures the classic cascade optimization problem.

## 7  Numerical Experiments

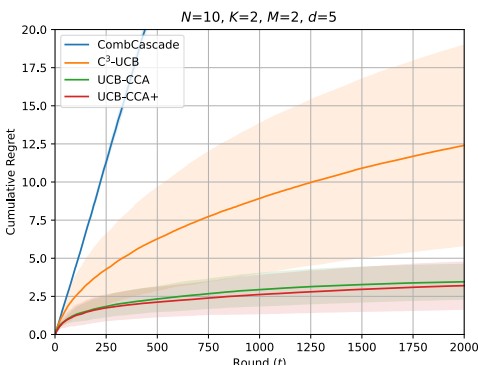
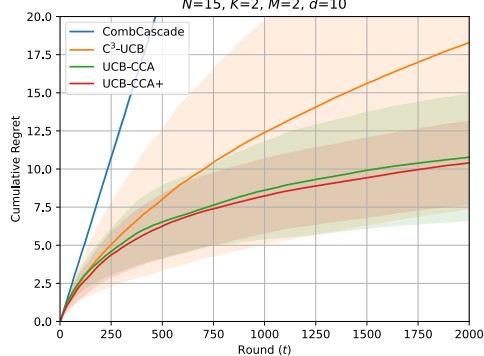

Figure 2: $N = 10$, $K = 2$, $M = 2$, and $d = 5$.         Figure 3: $N = 15$, $K = 2$, $M = 2$, and $d = 10$ .

In this section, we evaluate the performances of our proposed algorithms `UCB-CCA` and `UCB-CCA+` in numerical experiments and compare their performances with the existing combinatorial bandit algorithms `CombCascade` and `C³-UCB`. For simulations, we generate a random sample of the unknown time-invariant parameter $\theta^*$ from $\mathcal{N}(0, 1)$ at the beginning of the simulation. We sample $N$ feature vectors from $\mathcal{N}(0, 1)$ in each round $t$. At each round $t$, the oracle computes a sequence of assortments in decreasing order, forming a cascade $S_t$ based on given $w_t$. We assess the cumulative regret of `UCB-CCA`, `UCB-CCA+`, `CombCascade`, and `C³-UCB`. Note that a user's choice for `UCB-CCA` and `UCB-CCA+` is determined by the MNL logit choice model, whereas `C³-UCB` utilizes a linear model and `CombCascade` is a non-contextual model. Figure 2 and Figure 3 indicate that both `UCB-CCA` and `UCB-CCA+` significantly outperform `C³-UCB` and `CombCascade`. We also observe that `UCB-CCA+` shows a slight performance advantage over `UCB-CCA`, although the difference is not statistically significant. While `UCB-CCA+` has a sharper worst-case regret guarantee, `UCB-CCA` can provide favorable practical performances, with simpler implementation and computational efficiency.

## Acknowledgments and Disclosure of Funding

This work was supported by the National Research Foundation of Korea(NRF) grant funded by the Korea government(MSIT) (RS-2023-00222663, No. 2022R1C1C1006859, No. 2022R1A4A103057912, No. 2021M3E5D2A01024795) and by Creative-Pioneering Researchers Program through Seoul National University and by Naver.

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
