# A   More Discussions on the Related Work

Table 1 summarizes comparisons between our work and the previous works that propose algorithms in various combinatorial bandit settings: cascading bandits and assortment bandits.

For cascading bandits, there are two major objectives. One is called a *disjunctive* objective where the agent receives a positive reward when at least one item in the recommended sequence of items $K$ is attractive. The other one is a *conjunctive* objective where the agent receives a positive reward when all the items are attractive. Kveton et al. [13] first introduced the multi-armed cascading bandits with disjunctive objective, and Kveton et al. [14] proposed the cascading bandits with conjunctive objective. There is another difference between previous studies with these two objectives, which is the definition of the feasible set. The feasible set is an arbitrary subset of ground items in Kveton et al. [14], whereas it is a uniform matroid in Kveton et al. [13]. These previous studies propose UCB-type algorithms and derive both gap-dependent and gap-independent regret upper bounds.

Li et al. [18] generalize the above two models with contextual information. In this model, each item has its own weight that represents its attractiveness to the user. The weight of an item is assumed to have a linear relation with the feature of the item. The agent learns the unknown parameter from the feedback while maximizing the reward function for cascades. Li et al. [18] propose a UCB-type algorithm, referred as $C^3$-UCB, and prove that the $T$-step regret of $C^3$-UCB is upper bounded by $\tilde{\mathcal{O}}(d\sqrt{TK})$ where $d$ is the dimension of the contextual information vector, $K$ is the length of cascade, and $p^*$ is the minimum probability that the user examines all the items in the offered cascade for any cascade and any round. The regret bound studied in the aforementioned cascading bandit literature [13; 14; 18] is dependent on the length of the cascade ($K$) which is counter-intuitive results. The latest work by Vial et al. [25] removes the dependence on $K$ from their $T$-step regret upper bound, i.e. $\tilde{\mathcal{O}}(\sqrt{TN})$ where $N$ is the number of ground items, in the tabular case where there is no assumption on the structure of the weight. In the linear case, however, the $T$-step regret upper bound of their algorithm, referred as `CascadeWOFUL`, is $\tilde{\mathcal{O}}(\sqrt{Td(d+K)})$ which still scales with $K$.

There are many recent works on assortment bandits [4; 5; 8; 9; 22; 23; 7] using the multinomial choice model While, in the cascade bandits, the agent offers a cascade and a user examines it one by one, in assortment bandits, a user receives an assortment from the agent and examines all the items in a given assortment at once. Due to this difference, the agent receives feedback of only examined items in cascade bandits, but of all items in assortment bandits.

Agrawal et al. [4] and Agrawal et al. [5] propose Thompson sampling and UCB-type algorithms, respectively, in a non-contextual setting. Both show that their regret upper-bound is $\tilde{\mathcal{O}}(\sqrt{NT})$. Oh and Iyengar [22, 23] and Chen et al. [7] incorporate the contextual information into the MNL assortment bandits. They introduce the unknown time-invariant learning parameter which represents the utility of the item, i.e. $x_{ti}^\top \theta$ where $x_{ti}$ is the contextual vector of item $i$ at round $t$ and $\theta$ is the parameter that the agent is learning from the feedback. Oh and Iyengar [22] propose Thompson Sampling algorithm and derive $\tilde{\mathcal{O}}(d\sqrt{T})$. Oh and Iyengar [23] propose UCB-type algorithm, referred as `UCB-MNL`, and get a same regret bound $\tilde{\mathcal{O}}(d\sqrt{T})$.

# B   Proof of Theorem 4.3: $\alpha$-Regret Analysis

Consider weights $w, w' \in [0,1]^{[N]}$. We denote $w \geq w'$ if $w_i \geq w'_i$ holds for all $i \in [N]$.

**Lemma B.1.** *Given weights $w, w' \in [0,1]^{[N]}$ such that $w \geq w'$, we have that $f(S_t, w)$ is increasing with respect to $w$, that is, if $w \geq w'$, then for any $S_t \in \mathcal{S}$, it holds that $f(S_t, w) \geq f(S_t, w')$.*

*Proof.* It is easy to prove because of the structure of the expected reward fuction. we know that

$$f(S_t, w) = 1 - \prod_{k=1}^{|S_t|} p_t(i_0 | A_{tk}, w).$$

If $w$ increases, then $p_t(i_0 | A_{tk}, w)$ decrease and $f(S_t, w)$ increases.  □

**Lemma B.2.** *(Restatement of Lemma 4.4) Suppose $u_{ti} \geq w_{ti}^*$ for all $i \in [N]$. Then,*

$$f(S_t, u_t) - f(S_t, w_t^*) \leq \left(\frac{K}{K+1}\right)^{K+1} \max_{A_{tk} \in S_t} \max_{i \in A_{tk}} (u_{ti} - w_{ti}^*). \tag{7}$$

*Proof.* By the mean value theorem,

$$f(S_t, u_t) - f(S_t, w_t^*) = \nabla_\theta f(S_t, \bar{w})(\theta_t - \theta^*)$$

$$= \left\{ \prod_{A_{t\dot{k}} \in S_t} p_t(i_0|A_{t\dot{k}}, \bar{w}) \right\} \sum_{A_{tk} \in S_t} \sum_{i \in A_{tk}} p_t(i|A_{tk}, \bar{w}) x_{ti}^\top (\theta_t - \theta^*)$$

$$\leq \left(\frac{K}{K+1}\right)^{K+1} \max_{A_{tk} \in S_t} \max_{i \in A_{tk}} (u_{ti} - w_{ti}^*)$$

We can simplify $\left\{ \prod_{A_{t\dot{k}} \in S_t} p_t(i_0|A_{t\dot{k}}, \bar{w}) \right\} \sum_{A_{tk} \in S_t} \sum_{i \in A_{tk}} p_t(i|A_{tk}, \bar{w})$ on the second equality as follows if we denote $P_{tk} := \sum_{i \in A_{tk}} p_t(i|A_{tk}, \bar{w})$ for convenience:

$$\prod_{\dot{k} \in [K]} (1 - P_{t\dot{k}}) \sum_{k \in [K]} P_{tk}.$$

We can easily see that this expression is maximized as $\left(\frac{K}{K+1}\right)^{K+1}$ when $P_{tk} = \frac{1}{K+1}$ for all $k \in [K]$, since $0 < P_{tk} < 1$. $\square$

**Lemma B.3** (Lemma 4 in Oh and Iyengar [22]). *Let $\beta_t(\delta) = \frac{1}{2\kappa}\sqrt{d\log\left(1 + \frac{tKM}{d\lambda}\right) + 2\log\frac{1}{\delta}} + \frac{\sqrt{\lambda}}{\kappa}$. Then for any $\delta \geq 0$, we have*

$$\|\hat{\theta}_t - \theta^*\|_{V_t} \leq \beta_t(\delta)$$

*with a probability at least $1 - \delta$ for all round $t$.*

*Proof.* We adapt the proof of Lemma 4 in Oh and Iyengar [22] to our setting. We first denote the probability that a customer clicks the item $i$ in $A_{tk}$ as below:

$$p_t(i|A_{tk}, \theta) = \frac{\exp\left(x_{ti}^\top \theta\right)}{1 + \sum_{j \in A_{tk}} \exp\left(x_{tj}^\top \theta\right)}$$

Then, we define the function $G_t(\theta)$ (where $\theta$ is a parameter),

$$G_t(\theta) := \sum_{\tau=1}^{t} \sum_{k=1}^{O_\tau} \sum_{i \in A_{\tau k}} [(p_\tau(i|A_{\tau k}, \theta) - p_\tau(i|A_{\tau k}, \theta^*)) x_{\tau i}] + \lambda(\theta - \theta^*) \tag{8}$$

$G_t(\theta)$ represent the difference in the gradients of the ridge penalized maximum likelihood evaluated at $\theta$ and at $\theta^*$. Note that $\hat{\theta}$ can be obtained by minimizing Eq.(2). Therefore, the following equation is satisfied.

$$\sum_{\tau=1}^{t} \sum_{k=1}^{O_\tau} \sum_{i_m \in A_{\tau k}} \left(p_\tau(i_m|A_{\tau k}, \hat{\theta}) - y_{\tau km}\right) x_{\tau i_m} + \lambda\hat{\theta} = 0 \tag{9}$$

Now we have

$$G_t(\hat{\theta}) = \sum_{\tau=1}^{t} \sum_{k=1}^{O_\tau} \sum_{i \in A_{\tau k}} \left[\left(p_\tau(i|A_{\tau k}, \hat{\theta}) - p_\tau(i|A_{\tau k}, \theta^*)\right) x_{\tau i}\right] + \lambda\left(\hat{\theta} - \theta^*\right)$$

$$= \sum_{\tau=1}^{t} \sum_{k=1}^{O_\tau} \sum_{i_m \in A_{\tau k}} \left(p_\tau(i_m|A_{\tau k}, \hat{\theta}) - y_{\tau km}\right) x_{\tau i_m} + \lambda\hat{\theta}$$

$$+ \sum_{\tau=1}^{t} \sum_{k=1}^{O_\tau} \sum_{i_m \in A_{\tau k}} (p_\tau(i_m|A_{\tau k}, \theta^*) - y_{\tau km}) x_{\tau i_m} - \lambda\theta^*$$

$$= 0 + \sum_{\tau=1}^{t} \sum_{k=1}^{O_\tau} \sum_{i_m \in A_{\tau k}} \epsilon_{\tau km} x_{\tau i_m} - \lambda\theta^*$$

where the last equality is from Eq.(9) and the definition of $\epsilon_{tkm} = y_{tkm} - p_t(i_m|A_{tk}, \theta^*)$. For convenience, we define $Z_t := \sum_{\tau=1}^{t} \sum_{k=1}^{O_\tau} \sum_{i_m \in A_{\tau k}} \epsilon_{\tau km} x_{\tau i_m}$.

For any parameters $\theta_1, \theta_2 \in \mathbb{R}^d$, by mean value theorem, there exists $\bar{\theta} = c\theta_1 + (1-c)\theta_2$ with some $c \in (0, 1)$ such that

$$G_t(\theta_1) - G_t(\theta_2)$$

$$= \sum_{\tau=1}^{t} \sum_{k=1}^{O_\tau} \sum_{i \in A_{\tau k}} [(p_\tau(i|A_{\tau k}, \theta_1) - p_\tau(i|A_{\tau k}, \theta_2)) x_{\tau i}] + \lambda(\theta_1 - \theta_2)$$

$$= \left[ \left( \sum_{\tau=1}^{t} \sum_{k=1}^{O_\tau} \sum_{i \in A_{\tau k}} \sum_{j \in A_{\tau k}} \nabla_j p_\tau(i|A_{\tau k}, \bar{\theta}) x_{\tau i} x_{\tau j}^\top \right) + \lambda I_d \right] + \lambda(\theta_1 - \theta_2)$$

$$= \left[ \sum_{\tau=1}^{t} \sum_{k=1}^{O_\tau} \left( \sum_{i \in A_{\tau k}} p_\tau(i|A_{\tau k}, \bar{\theta}) x_{\tau i} x_{\tau i}^\top - \sum_{i,j \in A_{\tau k}} p_\tau(i|A_{\tau k}, \bar{\theta}) p_\tau(j|A_{\tau k}, \bar{\theta}) x_{\tau i} x_{\tau j}^\top \right) \right] (\theta_1 - \theta_2)$$

$$+ \lambda I_d(\theta_1 - \theta_2)$$

where $I_d$ is a $d \times d$ identity matrix. We define the matrix $H_{\tau k}$ as

$$H_{\tau k} := \sum_{i \in A_{\tau k}} p_\tau(i|A_{\tau k}, \bar{\theta}) x_{\tau i} x_{\tau i}^\top - \sum_{i \in A_{\tau k}} \sum_{j \in A_{\tau k}} p_\tau(i|A_{\tau k}, \bar{\theta}) p_\tau(j|A_{\tau k}, \bar{\theta}) x_{\tau i} x_{\tau j}^\top$$

We can see that $H_{\tau k}$ is positive semi-definite, since $H_{\tau k}$ is a Hessian of a negative log-likelihood which is convex.

Also, notice that

$$(x_i - x_j)(x_i - x_j)^\top = x_i x_i^\top + x_j x_j^\top - x_i x_j^\top - x_j x_i^\top \succeq 0$$

which implies $x_i x_i^\top + x_j x_j^\top \succeq x_i x_j^\top + x_j x_i^\top$. Therefore, it follows that

$$H_{\tau k} = \sum_{i \in A_{\tau k}} p_\tau(i|A_{\tau k}, \bar{\theta}) x_{\tau i} x_{\tau i}^\top - \sum_{i \in A_{\tau k}} \sum_{j \in A_{\tau k}} p_\tau(i|A_{\tau k}, \bar{\theta}) p_\tau(j|A_{\tau k}, \bar{\theta}) x_{\tau i} x_{\tau j}^\top$$

$$= \sum_{i \in A_{\tau k}} p_\tau(i|A_{\tau k}, \bar{\theta}) x_{\tau i} x_{\tau i}^\top - \frac{1}{2} \sum_{i \in A_{\tau k}} \sum_{j \in A_{\tau k}} p_\tau(i|A_{\tau k}, \bar{\theta}) p_\tau(j|A_{\tau k}, \bar{\theta}) \left( x_{\tau i} x_{\tau j}^\top + x_{\tau j} x_{\tau i}^\top \right)$$

$$\succeq \sum_{i \in A_{\tau k}} p_\tau(i|A_{\tau k}, \bar{\theta}) x_{\tau i} x_{\tau i}^\top - \frac{1}{2} \sum_{i \in A_{\tau k}} \sum_{j \in A_{\tau k}} p_\tau(i|A_{\tau k}, \bar{\theta}) p_\tau(j|A_{\tau k}, \bar{\theta}) \left( x_{\tau i} x_{\tau i}^\top + x_{\tau j} x_{\tau j}^\top \right)$$

$$= \sum_{i \in A_{\tau k}} p_\tau(i|A_{\tau k}, \bar{\theta}) x_{\tau i} x_{\tau i}^\top - \sum_{i \in A_{\tau k}} \sum_{j \in A_{\tau k}} p_\tau(i|A_{\tau k}, \bar{\theta}) p_\tau(j|A_{\tau k}, \bar{\theta}) x_{\tau i} x_{\tau i}^\top$$

$$= \sum_{i \in A_{\tau k}} p_\tau(i|A_{\tau k}, \bar{\theta}) \left( 1 - \sum_{j \in A_{\tau k}} p_\tau(j|A_{\tau k}, \bar{\theta}) \right) x_{\tau i} x_{\tau i}^\top$$

$$= \sum_{i \in A_{\tau k}} p_\tau(i|A_{\tau k}, \bar{\theta}) p_\tau(i_0|A_{\tau k}, \bar{\theta}) x_{\tau i} x_{\tau i}^\top$$

where $p_\tau(i_0|A_{\tau k}, \theta) := \frac{1}{1 + \sum_{j \in A_{\tau k}} x_{\tau i}^\top \theta}$ is the click probability of the outside option with respect to $\theta$ at round $\tau$. Now,

$$G_t(\theta_1) - G_t(\theta_2) = \left[ \sum_{\tau=1}^{t} \sum_{k=1}^{O_\tau} H_{\tau k} + \lambda I_d \right] (\theta_1 - \theta_2)$$

$$\geq \left[ \sum_{\tau=1}^{t} \sum_{k=1}^{O_\tau} \sum_{i \in A_{\tau k}} p_\tau(i|A_{\tau k}, \bar{\theta}) p_\tau(i_0|A_{\tau k}, \bar{\theta}) x_{\tau i} x_{\tau i}^\top + \lambda I_d \right] (\theta_1 - \theta_2)$$

$$:= \mathcal{H}(\bar{\theta})(\theta_1 - \theta_2).$$

$p_\tau(i|A_{\tau k}, \bar{\theta})p_\tau(i_0|A_{\tau k}, \bar{\theta})$ is lower-bounded by $\kappa$ from Assumption 4.2. Then we have

$$(\theta_1 - \theta_2)^\top (G_t(\theta_1) - G_t(\theta_2)) \geq (\theta_1 - \theta_2)^\top (\kappa V_t)(\theta_1 - \theta_2) > 0$$

for any $\theta_1 \neq \theta_2$. . From the definition on Eq.(8), $G_t(\theta^*) = 0$. Hence, for any $\theta \in \mathbb{R}^d$, we have

$$\begin{aligned}
||G_t(\theta)||^2_{V_t^{-1}} &= ||G_t(\theta) - G_t(\theta^*)||^2_{V_t^{-1}} \\
&= (G_t(\theta) - G_t(\theta^*))^\top V_t^{-1}(G_t(\theta) - G_t(\theta^*)) \\
&\geq (\theta - \theta^*)^\top \mathcal{H}(\bar{\theta})V_t^{-1}\mathcal{H}(\bar{\theta})(\theta - \theta^*) \\
&\geq \kappa^2 (\theta - \theta^*)^\top V_t(\theta - \theta^*) \\
&= \kappa^2 ||\theta - \theta^*||^2_{V_t}
\end{aligned}$$

where the last inequality is from $\mathcal{H}(\bar{\theta}) \succeq \kappa V_t$. Using $G_t(\hat{\theta}) = Z_t - \lambda\theta^*$, we have

$$\kappa||\hat{\theta} - \theta^*||_{V_t} \leq ||G_t(\hat{\theta})||_{V_t^{-1}} \leq ||Z_t||_{V_t^{-1}} + \lambda||\theta^*||_{V_t^{-1}}$$

Recall that $Z_t = \sum_{\tau=1}^t \sum_{k \in O_\tau} \sum_{i_m \in A_{\tau k}} \epsilon_{\tau km} x_{\tau i_m}$ and $\epsilon_{\tau km}$ is sub-Gaussian with parameter $\sigma$, then we can apply Theorem 1 in Abbasi-Yadkori et al. [1]:

$$||Z_t||^2_{V_t^{-1}} \leq 2\sigma^2 \log\left(\frac{\det(V_t)^{1/2}\det(V)^{-1/2}}{\delta}\right)$$

with probability at least $1 - \delta$. Then we combine with Lemma B.9:

$$\begin{aligned}
||Z_t||^2_{V_t^{-1}} &\leq 2\sigma^2 \left[\frac{d}{2}\ln\left(\frac{\text{trace}V + tKM}{d}\right) - \frac{1}{2}\ln\det V + \ln\frac{1}{\delta}\right] \\
&\leq 2\sigma^2 \left[\frac{d}{2}\ln(d\lambda + tKM)\, d - \frac{1}{2}\ln\lambda^d + \ln\frac{1}{\delta}\right] \\
&= 2\sigma^2 \left[\frac{d}{2}\ln\left(\lambda + \frac{tKM}{d}\right) - \frac{d}{2}\ln\lambda + \ln\frac{1}{\delta}\right] \\
&= 2\sigma^2 \left[\frac{d}{2}\ln\left(1 + \frac{tKM}{d\lambda}\right) + \ln\frac{1}{\delta}\right]
\end{aligned}$$

where the first inequality is from the fact that $V = \lambda I$. Next we need to bound $\lambda||\theta^*||_{V_t^{-1}}$. We have

$$||\theta^*||^2_{V_t^{-1}} \leq \frac{||\theta^*||^2_2}{\lambda_{\min}(V_t)} \leq \frac{||\theta^*||^2_2}{\lambda_{\min}(V)} \leq \frac{||\theta^*||^2_2}{\lambda}.$$

By Assumption 4.1 that $||\theta^*||^2_2 \leq 1$, $\lambda||\theta^*||_{V_t^{-1}} \leq \sqrt{\lambda}$. Recall that $\sigma = \frac{1}{2}$ in our problem. Combining the bound of $||Z_t||_{V_t^{-1}}$ and $\lambda||\theta^*||_{V_t^{-1}}$, we have

$$||\hat{\theta}_t - \theta^*||_{V_t} \leq \frac{1}{2\kappa}\sqrt{d\ln\left(1 + \frac{tKM}{d\lambda} + 2\ln\frac{1}{\delta}\right)} + \frac{\sqrt{\lambda}}{\kappa}$$

with probability at least $1 - \delta$. $\qquad\square$

Thus $\hat{\theta}_t$ lies in the ellipsoid centered at $\theta^*$ with confidence radius $\beta_t(\delta)$ under $V_t$ norm. Building on this, we can define an upper confidence bound of the true weight for each base arm $i$ by

$$u_{ti} = x_{ti}^\top \hat{\theta}_{t-1} + \beta_{t-1}||x_{ti}||_{V_{t-1}^{-1}}$$

Recall that we define the high-probability concentration event $\mathcal{E}_t(\delta) := \{||\hat{\theta}_t - \theta^*||_{V_t} \leq \beta_t(\delta)\}$. The fact that $u_{ti}$ is an upper confidence bound of true weight $w_{ti}^* = x_{ti}^\top \theta^*$ is proved in the following Lemma B.4 and Lemma B.5.

**Lemma B.4** (Optimism). *On event $\mathcal{E}_t(\delta)$, for every item $i \in [N]$, we have*

$$u_{ti} \geq w_{ti}^*. \tag{10}$$

*Proof.* Recall that $w_{ti}^* = x_{ti}^\top \theta^*$. By Hölder's inequality,

$$
\begin{aligned}
\left| x_{ti}^\top \hat{\theta}_{t-1} - x_{ti}^\top \theta^* \right| &= \left| \left[ V_{t-1}^{1/2} \left( \hat{\theta}_{t-1} - \theta^* \right) \right]^\top \left( V_{t-1}^{-1/2} x_{ti} \right) \right| \\
&\leq \| V_{t-1}^{1/2} \left( \hat{\theta}_{t-1} - \theta^* \right) \|_2 \| V_{t-1}^{-1/2} x_{ti} \|_2 \\
&= \| \hat{\theta}_{t-1} - \theta^* \|_{V_{t-1}} \| x_{ti} \|_{V_{t-1}^{-1}} \\
&\leq \beta_{t-1}(\delta) \| x_{ti} \|_{V_{t-1}^{-1}}.
\end{aligned} \tag{11}
$$

From the last inequality above, we have

$$-\beta_{t-1}(\delta) \| x_{ti} \|_{V_{t-1}^{-1}} \leq x_{ti}^\top \hat{\theta}_{t-1} - x_{ti}^\top \theta^*.$$

Add $\beta_{t-1}(\delta) \| x_{ti} \|_{V_{t-1}^{-1}}$, then we have

$$0 \leq \left( x_{ti}^\top \hat{\theta}_{t-1} + \beta_{t-1}(\delta) \| x_{ti} \|_{V_{t-1}^{-1}} \right) - x_{ti}^\top \theta^* = u_{ti} - w_{ti}^*.$$

$\square$

**Lemma B.5** (Concentration of UCB weights). *On event $\mathcal{E}_t(\delta)$, for every item $i \in [N]$, we have*

$$u_{ti} - w_{ti}^* \leq 2\beta_{t-1}(\delta) \| x_{ti} \|_{V_{t-1}^{-1}}. \tag{12}$$

*Proof.* From inequality (11), we have

$$\beta_{t-1}(\delta) \| x_{ti} \|_{V_{t-1}^{-1}} \geq x_{ti}^\top \hat{\theta}_{t-1} - x_{ti}^\top \theta^*.$$

Adding $\beta_{t-1}(\delta) \| x_{ti} \|_{V_{t-1}^{-1}}$ to both sides gives,

$$2\beta_{t-1}(\delta) \| x_{ti} \|_{V_{t-1}^{-1}} \geq \left( x_{ti}^\top \hat{\theta}_{t-1} + \beta_{t-1}(\delta) \| x_{ti} \|_{V_{t-1}^{-1}} \right) - x_{ti}^\top \theta^* = u_{ti} - w_{ti}^*$$

$\square$

**Lemma B.6.** *For any round $t$ and action $S_t$, we have*

$$\mathcal{R}^\alpha(t, S_t) \leq 2 \left( \frac{K}{K+1} \right)^{K+1} \max_{k \in [|S_t|]} \max_{i \in A_{tk}} \beta_{t-1}(\delta) \| x_{ti} \|_{V_{t-1}^{-1}}$$

*Proof.* Let $S^{u_t} = \arg \max_{S \in \mathcal{S}} f(S, u_t)$ and recall that $S_t^* = \arg \max_{S \in \mathcal{S}} f(S, w_t^*)$. Then

$$f(S_t, u_t) \geq \alpha f(S^{u_t}, u_t) \geq \alpha f(S_t^*, u_t) \geq \alpha f(S_t^*, w_t^*).$$

The first inequality is by the definition of $\alpha$-approximate oracle. The second inequality comes from the fact that $S^{u_t}$ has the maximum expected reward when $u_t$ is given. Combining Lemma B.4 which states $u_t \geq w_t^*$ and Lemma B.1 which is about the monotonicity of the expected reward function, we can obtain the last inequality. Then we can bound the $\mathcal{R}^\alpha(t, S_t)$ with the expected reward difference of $S_t$ between given $u_t$ and $w_t^*$ as follows:

$$\mathcal{R}^\alpha(t, S_t) = \alpha f(S_t^*, w_t^*) - f(S_t, w_t^*) \leq f(S_t, u_t) - f(S_t, w_t^*).$$

By Lemma B.2 and Lemma B.5,

$$
\begin{aligned}
\mathcal{R}^\alpha(t, S_t) &\leq f(S_t, u_t) - f(S_t, w_t^*) \\
&\leq \left( \frac{K}{K+1} \right)^{K+1} \max_{k \in [|S_t|]} \max_{i \in A_{tk}} (u_{ti} - w_{ti}^*) \\
&\leq 2 \left( \frac{K}{K+1} \right)^{K+1} \max_{k \in [|S_t|]} \max_{i \in A_{tk}} \beta_{t-1}(\delta) \| x_{ti} \|_{V_{t-1}^{-1}}
\end{aligned}
$$

$\square$

**Lemma B.7.** *Let $x_i \in \mathbb{R}^d$ and $I \in \mathbb{R}^{d \times d}$ be an identity matrix. Then we have*

$$\det\left(I + \sum_{i=1}^n x_i x_i^\top\right) \geq 1 + \sum_{i=1}^n \|x_i\|_2^2.$$

*Proof.* Let the eigenvalues of $\sum_{i=1}^n x_i x_i^\top$ be $\lambda_1, \ldots, \lambda_d$ where $\lambda_j \geq 0$ for all $1 \leq j \leq d$. Then we have

$$\det\left(I + \sum_{i=1}^n x_i x_i^\top\right) = \prod_{j=1}^d (1 + \lambda_j) \geq 1 + \sum_{j=1}^d \lambda_j = 1 - d + \sum_{j=1}^d (1 + \lambda_j)$$

$$= 1 - d + \text{trace}\left(I + \sum_{i=1}^n x_i x_i^\top\right) = 1 - d + d + \sum_{i=1}^n \|x_i\|_2^2$$

$$= 1 + \sum_{i=1}^n \|x_i\|_2^2$$

$\square$

**Lemma B.8** (Maximal elliptical potential).

$$\sum_{\tau=1}^t \max_{k \in [O_\tau]} \max_{i \in A_{\tau k}} \|x_{\tau i}\|_{V_\tau^{-1}}^2 \leq 2 \ln\left(\frac{\det(V_t)}{\lambda^d}\right)$$

*Proof.* Let $\lambda_{\min}(V_\tau)$ be the minimum eigenvalue of $V_\tau$. Since $\lambda \geq 1$ and $\|x_{\tau i}\|_{V_{\tau-1}}^2 \leq \frac{\|x_{\tau i}\|_2^2}{\lambda_{\min}(V_{\tau-1})} \leq \frac{1}{\lambda}$, we have

$$\max_{k \in [O_\tau]} \max_{i \in A_{\tau k}} \|x_{\tau i}\|_{V_{\tau-1}^{-1}}^2 \leq 1.$$

Using the fact that $z \leq 2\ln(1 + z)$ for any $z \in [0, 1]$, we have

$$\sum_{\tau=1}^t \max_{k \in [O_\tau]} \max_{i \in A_{\tau k}} \|x_{\tau i}\|_{V_\tau^{-1}}^2 \leq 2 \sum_{\tau=1}^t \ln\left(1 + \max_{k \in [O_\tau]} \max_{i \in A_{\tau k}} \|x_{\tau i}\|_{V_\tau^{-1}}^2\right)$$

$$= 2 \ln \prod_{\tau=1}^t \left(1 + \max_{k \in [O_\tau]} \max_{i \in A_{\tau k}} \|x_{\tau i}\|_{V_\tau^{-1}}^2\right). \tag{13}$$

Now we upper bound $\prod_{\tau=1}^t \left(1 + \max_{k \in [O_\tau]} \max_{i \in A_{\tau k}} \|x_{\tau i}\|_{V_\tau^{-1}}^2\right)$ from $\det(V_t)$.

$$\det(V_t) = \det\left(V_{t-1} + \sum_{k=1}^{O_t} \sum_{i \in A_{tk}} x_{ti} x_{ti}^\top\right)$$

$$= \det(V_{t-1}) \det\left(I + V_{t-1}^{-1/2} \sum_{k=1}^{O_t} \sum_{i \in A_{tk}} x_{ti} x_{ti}^\top V_{t-1}^{-1/2}\right)$$

$$= \det(V_{t-1}) \det\left(I + \sum_{k=1}^{O_t} \sum_{i \in A_{tk}} \left(V_{t-1}^{-1/2} x_{ti}\right) \left(V_{t-1}^{-1/2} x_{ti}\right)^\top\right)$$

$$\geq \det(V_{t-1}) \left(1 + \sum_{k=1}^{O_t} \sum_{i \in A_{tk}} \|x_{ti}\|_{V_{t-1}^{-1}}^2\right)$$

$$\geq \det(\lambda I) \prod_{\tau=1}^t \left(1 + \sum_{k=1}^{O_\tau} \sum_{i \in A_{\tau k}} \|x_{\tau i}\|_{V_{\tau-1}^{-1}}^2\right)$$

$$\geq \det(\lambda I) \prod_{\tau=1}^t \left(1 + \max_{k \in [O_\tau]} \max_{i \in A_{\tau k}} \|x_{\tau i}\|_{V_{\tau-1}^{-1}}^2\right).$$

The second equality above is from the fact that $V + U = V^{1/2}(I + V^{-1/2}UV^{-1/2})V^{1/2}$ for a symmetric positive definite matrix $V$. The first inequality above can be obtained by applying Lemma B.7. Applying the first inequality repeatedly, we can get the second inequality above. Thus, we have

$$\prod_{\tau=1}^{t}\left(1 + \max_{k\in[O_\tau]}\max_{i\in A_{\tau k}}||x_{\tau i}||^2_{V_{\tau-1}^{-1}}\right) \leq \frac{\det(V_t)}{\det(\lambda I)}. \tag{14}$$

Then applying Eq.(14) to Eq.(13), we complete the proof as follows:

$$\sum_{\tau=1}^{t}\max_{k\in[O_\tau]}\max_{i\in A_{\tau k}}||x_{\tau i}||^2_{V_\tau^{-1}} \leq 2\ln\frac{\det(V_t)}{\det(\lambda I)} \leq 2\ln\frac{\det(V_t)}{\lambda^d}$$

where the last inequality is from Lemma B.9. $\qquad\square$

**Lemma B.9** (Lemma 10 in Abbasi-Yadkori et al. [1]). *$det\,(V_t)$ is increasing with respect to $t$ and*

$$det\,(V_t) \leq \left(\lambda + \frac{tKM}{d}\right)^d$$

*Proof.* We first prove that $\det(V_t)$ is increasing with respect to $t$. For any symmetric positive definite matrix $\tilde{V} \in \mathbb{R}^{d\times d}$ and column vector $x \in \mathbb{R}^{d\times 1}$, we can see $\det(\tilde{V} + xx^\top) \geq \det(\tilde{V})$ as follows:

$$\begin{aligned}\det(\tilde{V} + xx^\top) &= \det(\tilde{V})\det(I + \tilde{V}^{-1/2}xx^\top\tilde{V}^{-1/2}) \\ &= \det(\tilde{V})\det(1 + ||\tilde{V}^{-1/2}x||^2) \\ &\geq \det(\tilde{V}).\end{aligned}$$

The second equality above is due to Sylvester's determinant theorem, which states that $\det(I + AB) = \det(I + BA)$.

Next, we prove the inequality in Lemma B.9. Let $\lambda_1, \ldots, \lambda_d$ be the eigenvalues of $V_t \in \mathbb{R}^{d\times d}$. Then

$$\begin{aligned}\det(V_t) &= \lambda_1\lambda_2\cdots\lambda_d \\ &\leq \left(\frac{\lambda_1 + \cdots + \lambda_d}{d}\right)^d = \left(\frac{\mathrm{trace}(V_t)}{d}\right)^d.\end{aligned}$$

The second inequality above is from the AM-GM inequality. Now we need to bound $\mathrm{trace}(V_t)$ as follows:

$$\begin{aligned}\mathrm{trace}(V_t) &= \mathrm{trace}(V) + \sum_{\tau=1}^{t}\sum_{k=1}^{O_\tau}\sum_{i\in A_{\tau k}}\mathrm{trace}(x_{\tau i}x_{\tau i}^\top) \\ &= d\lambda + \sum_{\tau=1}^{t}\sum_{k=1}^{O_\tau}\sum_{i\in A_{\tau k}}||x_{\tau i}||^2_2 \\ &\leq d\lambda + tKM\end{aligned}$$

The second inequality is due to Assumption 4.1 that $||x_{ti}|| \leq 1$. Thus, $\det(V_t) \leq \left(\lambda + \frac{tKM}{d}\right)^d$. $\quad\square$

Now, we can prove Theorem 4.3.

**Proof of Theorem 4.3.** Suppose B.3 holds for all round $t$. Then, with probability $1 - \delta$, we have

$$\mathcal{R}^\alpha(T) = \mathbb{E}\left[\sum_{t=1}^{T} \mathcal{R}^\alpha(t, S_t)\right]$$

$$\leq \mathbb{E}\left[\sum_{t=1}^{T} 2\left(\frac{K}{K+1}\right)^{K+1} \max_{k\in[|S_t|]} \max_{i\in A_{tk}} \beta_{t-1}(\delta)||x_{ti}||_{V_{t-1}^{-1}}\right] \tag{15}$$

$$\leq \mathbb{E}\left[2\left(\frac{K}{K+1}\right)^{K+1} \beta_T(\delta) \sum_{t=1}^{T} \max_{k\in[|S_t|]} \max_{i\in A_{tk}} ||x_{ti}||_{V_{t-1}^{-1}}\right] \tag{16}$$

$$= \mathbb{E}\left[2\left(\frac{K}{K+1}\right)^{K+1} \beta_T(\delta) \sum_{t=1}^{T} \max_{k\in[O_t]} \max_{i\in A_{tk}} ||x_{ti}||_{V_{t-1}^{-1}}\right] \tag{17}$$

Eq.(15) comes from Lemma B.6. Using the fact that $\beta_t(\delta)$ is increasing with respect to $t$, Eq.(16) is satisfied. Note that the upper bound of $\mathcal{R}^\alpha(t, S_t)$ is in terms of all assortments of $S_t$ in Eq.(16). The previous work [18] mentioned that it is hard to estimate an upper bound for $2\max_{k\in[K]} \max_{i\in A_{tk}} \beta_{t-1}(\delta)||x_{ti}||_{V_{t-1}^{-1}}$ because $V_t$ only contains information of observed assortments. We cope with this by max operations and the property of UCB-CCA that the largest item in $S_t$ in terms of $||x_{ti}||_{V_{t-1}^{-1}}$ is always in the first assortment. Then, we have

$$\max_{k\in[|S_t|]} \max_{i\in A_{tk}} ||x_{ti}||_{V_{t-1}^{-1}} = \max_{k\in[O_t]} \max_{i\in A_{tk}} ||x_{ti}||_{V_{t-1}^{-1}}$$

which is stated in Eq.(5), and thus Eq.(17) is given by the equation above.

We complete the remain part as follows:

$$\mathcal{R}^\alpha(T) \leq \mathbb{E}\left[2C_\kappa \beta_T(\delta) \sum_{t=1}^{T} \max_{k\in[O_t]} \max_{i\in A_{tk}} ||x_{ti}||_{V_{t-1}^{-1}}\right]$$

$$\leq \mathbb{E}\left[2C_\kappa \beta_T(\delta) \sqrt{\sum_{t=1}^{T} 1^2 \sum_{t=1}^{T} \max_{k\in[O_t]} \max_{i\in A_{tk}} ||x_{ti}||_{V_{t-1}^{-1}}^2}\right]$$

$$\leq \mathbb{E}\left[2C_\kappa \left(\frac{1}{2\kappa}\sqrt{d\log\left(1 + \frac{TKM}{d\lambda}\right) + 2\log\frac{1}{\delta}} + \frac{\sqrt{\lambda}}{\kappa}\right)\sqrt{T \cdot 2\ln\left(\frac{\det(V_t)}{\lambda^d}\right)}\right]$$

$$\leq 2\sqrt{2}C_\kappa \left(\frac{1}{2\kappa}\sqrt{d\log\left(1 + \frac{TKM}{d\lambda}\right) + 2\log\frac{1}{\delta}} + \frac{\sqrt{\lambda}}{\kappa}\right)\sqrt{Td\ln\left(1 + \frac{TKM}{d\lambda}\right)}.$$

The first inequality above is by applying Cauchy-Schwartz inequality. The second inequality comes from the definition of $\beta_t(\delta)$ in Lemma B.3 and Lemma B.7. The last inequality is from the upper bound of $\det(V_t)$ in Lemma B.9

Since we set $\delta = \frac{1}{t^2}$, we have

$$\mathcal{R}^\alpha(T) = \leq 2\sqrt{2}C_\kappa \left(\frac{1}{2\kappa}\sqrt{d\log\left(1 + \frac{TKM}{d\lambda}\right) + 4\log T} + \frac{\sqrt{\lambda}}{\kappa}\right)\sqrt{Td\ln\left(1 + \frac{TKM}{d\lambda}\right)}.$$

$\square$

## C  Proof of Theorem 5.2

Since UCB-CCA+ is a parametric-based algorithm, we redefine our notation for convenience. We denote the user click probability of $m$-th item in $A_{tk}$ and probability of the outside option in $A_{tk}$ as:

$$p_t(i_m|A_{tk}, \theta) = \frac{\exp\left(x_{ti_m}^\top \theta\right)}{1 + \sum_{j\in A_{tk}} \exp\left(x_{tj}^\top \theta\right)}.$$

We also denote the expected reward function as:

$$f(S_t, \theta^*) = \sum_{k=1}^{K} \left\{ \prod_{\hat{k}=1}^{k-1} p_t(i_0|A_{t\hat{k}}, \theta^*) \right\} \sum_{i \in A_{tk}} p_t(i|A_{tk}, \theta^*).$$

We define the first and second derivative of $p_t(i|A_{tk}, \theta)$ with respect to the arm $i$:

$$\dot{p}_t(i|A_{tk}, \theta) := p_t(i|A_{tk}, \theta)p_t(i_0|A_{tk}, \theta)$$
$$\ddot{p}_t(i|A_{tk}, \theta) := p_t(i|A_{tk}, \theta)p_t(i_0|A_{tk}, \theta)(1 - 2p_t(i_0|A_{tk}, \theta)).$$

Additionally, we define the new design matrix containing local information:

$$H_t(\theta) := \sum_{\tau=1}^{t} \sum_{k \in [O_\tau]} \sum_{i \in A_{\tau k}} \dot{p}_\tau(i|A_{\tau k}, \theta)x_{\tau i}x_{\tau i}^\top + \lambda_t I_d \tag{18}$$

$$H_t := \sum_{\tau=1}^{t} \sum_{k \in [O_\tau]} \sum_{i \in A_{\tau k}} \dot{p}_\tau(i|A_{\tau k}, \theta^*)x_{\tau i}x_{\tau i}^\top + \lambda_t I_d. \tag{19}$$

The negative log-likelihood function under any parameter $\theta$ is given by:

$$\ell_t(\theta) := -\sum_{\tau=1}^{t} \sum_{k=1}^{O_\tau} \sum_{i_m \in A_{\tau k}} y_{\tau k m} \log p_\tau(i_m|A_{\tau k}, \theta)$$

where $y_{\tau k m}$ is a user choice random variable on the $m$-th item $i_m$ in the $k$-th assortment $A_{\tau k}$ at round $\tau$. And the regularized maximum likelihood estimates $\hat{\theta}_t$ is given by minimizing the regularized log-likelihood function:

$$\hat{\theta}_t = \arg\min_\theta \left[ \ell_t(\theta) + \frac{\lambda_t}{2} ||\theta||_2^2 \right]$$

where the penalty parameter $\lambda_t \geq 1$. To obtain $\hat{\theta}_t$, we take the gradient of the above regularized log-likelihood function with respect to $\theta$:

$$\nabla_\theta \left[ \ell_t(\theta) + \frac{\lambda_t}{2} ||\theta||_2^2 \right] = \sum_{\tau=1}^{t} \sum_{k=1}^{O_\tau} \sum_{i \in A_{\tau k}} \{p_\tau(i|A_{\tau k}, \theta) - y_{\tau k m}\} x_{\tau i} + \lambda_t \theta. \tag{20}$$

And we denote the function $g_t(\theta)$ as follows:

$$g_t(\theta) := \sum_{\tau=1}^{t} \sum_{k=1}^{O_\tau} \sum_{i \in A_{\tau k}} p_\tau(i|A_{\tau k}, \theta)x_{\tau i} + \lambda_t \theta. \tag{21}$$

Since $\hat{\theta}_t$ is the minimizer of the regularized negative log-likelihood function, we can get $\hat{\theta}_t$ by setting Eq.(20) to 0. Then we can see that $g_t(\hat{\theta}_t)$ can be represented as a function of feature vector $x_{ti}$ and user choice variable $y_{tkm}$:

$$\sum_{\tau=1}^{t} \sum_{k=1}^{O_\tau} \sum_{i \in A_{\tau k}} y_{\tau k m} x_{\tau i} = \sum_{\tau=1}^{t} \sum_{k=1}^{O_\tau} \sum_{i \in A_{\tau k}} p_\tau(i|A_{\tau k}, \hat{\theta}_t) + \lambda_t \hat{\theta}_t x_{\tau i} = g_t(\hat{\theta}_t) \tag{22}$$

Now, the following lemma illustrates the relationship between the function $g_t(\theta)$ and its input parameter $\theta$.

**Lemma C.1.** *For any parameter $\theta_1, \theta_2 \in \mathbb{R}^d$, the following equality hold:*

$$g_t(\theta_1) - g_t(\theta_2) = \mathbb{G}_t(\theta_1, \theta_2)$$
$$||g_t(\theta_1) - g_t(\theta_2)||_{\mathbb{G}_t^{-1}(\theta_1, \theta_2)} = ||\theta_1 - \theta_2||_{\mathbb{G}_t(\theta_1, \theta_2)}$$

*where $\mathbb{G}_t(\theta_1, \theta_2) = \sum_{\tau=1}^{t} \sum_{k=1}^{O_\tau} \sum_{i \in A_{\tau k}} \left[ \int_{v=0}^{1} \dot{p}_\tau(i|A_{\tau k}, v\theta_2 + (1-v)\theta_1)dv \right] x_{\tau i}x_{\tau i}^\top + \lambda_t I_d$ and $I_d$ is $d$-dimensional identitiy matrix.*

*Proof.* We first derive the first equality in Lemma C.1.

$$g_t(\theta_1) - g_t(\theta_2) = \sum_{\tau=1}^{t}\sum_{k=1}^{O_\tau}\sum_{i\in A_{\tau k}} \{p_\tau(i|A_{\tau k},\theta_1) - p_\tau(i|A_{\tau k},\theta_2)\}\, x_{\tau i} + \lambda_t(\theta_1 - \theta_2)$$

$$= \sum_{\tau=1}^{t}\sum_{k=1}^{O_\tau}\sum_{i\in A_{\tau k}} \left[\int_{v=0}^{1}\dot{p}_\tau(i|A_{\tau k},v\theta_2 + (1-v)\theta_1)dv\right] x_{\tau i}x_{\tau i}^{\top}(\theta_1 - \theta_2) + \lambda_t(\theta_1 - \theta_2)$$

$$= \left(\sum_{\tau=1}^{t}\sum_{k=1}^{O_\tau}\sum_{i\in A_{\tau k}} \left[\int_{v=0}^{1}\dot{p}_\tau(i|A_{\tau k},v\theta_2 + (1-v)\theta_1)dv\right] x_{\tau i}x_{\tau i}^{\top} + \lambda_t I_d\right)(\theta_1 - \theta_2)$$

$$= \mathbb{G}_t(\theta_1,\theta_2)(\theta_1 - \theta_2)$$

Since $\int_{v=0}^{1}\dot{p}_t(i|A_{tk},v\theta_2 + (1-v)\theta_1)dv \geq \kappa$ from Assumption 4.2 and the definition of $\dot{p}_t(i|A_{tk},\theta)$, $\mathbb{G}_t(\theta_1,\theta_2)(\theta_1 - \theta_2)$ is positive definite. Thus, we can derive the second equality in Lemma C.1 from the first eqality in the same lemma. $\square$

**Theorem C.2** (Theorem 4 in Abeille et al. [2]). *Let $\{\mathcal{F}_t\}_{t=1}^{\infty}$ be a filtration. Let $\{x_t\}_{t=1}^{\infty}$ be a stochastic process in such that $x_t$ is $\mathcal{F}_t$ measurable. Let $\{\epsilon_t\}_{t=2}^{\infty}$ be a martingale difference sequence such that $\epsilon_{t+1}$ is $\mathcal{F}_{t+1}$ measurable. Furthermore, assume that conditionally on $\mathcal{F}_t$ we have $|\epsilon_{t+1}| \leq 1$ almost surely, and note $\sigma_t^2 := \mathbb{E}\left[\epsilon_{t+1}^2 \mid \mathcal{F}_t\right]$. Let $\{\lambda_t\}_{t=1}^{\infty}$ be a predictable sequence of non-negative scalars. Define: $\mathbb{H}_t := \sum_{\tau=1}^{t}\sigma_\tau^2 x_\tau x_\tau^2 + \lambda_t I_d$, $\mathbb{M}_t := \sum_{\tau=1}^{t}\epsilon_{\tau+1}x_\tau$ Then for any $\delta \in (0,1]$:*

$$\mathbb{P}\left(\exists t \geq 1, ||\mathbb{M}_t||_{\mathbb{H}_t^{-1}} \geq \frac{\sqrt{\lambda_t}}{2} + \frac{2}{\sqrt{\lambda_t}}\log\left(\frac{\det(\mathbb{H}_t)^{\frac{1}{2}}\lambda_t^{-\frac{d}{2}}}{\delta}\right) + \frac{2}{\sqrt{\lambda_t}}d\log 2\right) \leq \delta$$

Using Theorem C.2, we show that the following holds with high probability.

**Lemma C.3.** *With $\hat{\theta}_t$ as the regularized maximum log-likelihood estimate as defined in Eq.(2), the following inequality holds with probability at least $1 - \delta$:*

$$\forall t \geq 1, ||g_t(\hat{\theta}_t) - g_t(\theta^*)||_{H_t^{-1}} \leq \gamma_t(\delta), \tag{23}$$

*where the confidence radius $\gamma_t(\delta) := \frac{3\sqrt{\lambda_t}}{2} + \frac{2}{\sqrt{\lambda_t}}\log\left(\frac{(\lambda_t+tKM/d)^{d/2}\lambda_t^{-d/2}}{\delta}\right) + \frac{2d}{\sqrt{\lambda_t}}\log 2$.*

*Proof.* In Eq.(22), the following holds:

$$g_t(\hat{\theta}_t) = \sum_{\tau=1}^{t}\sum_{k=1}^{O_\tau}\sum_{i\in A_{\tau k}} y_{\tau km}x_{\tau i}.$$

Subtract $g_t(\theta^*) = \sum_{\tau=1}^{t-1}\sum_{k=1}^{O_\tau}\sum_{i_m\in A_{\tau k}} p_\tau(i_m|A_{\tau k},\theta^*) + \lambda_t\theta^*$ from both sides of the above equation, then we get:

$$g_t(\hat{\theta}_t) - g_t(\theta^*) = \sum_{\tau=1}^{t}\sum_{k=1}^{O_\tau}\sum_{i_m\in A_{\tau k}} \{y_{\tau km} - p_\tau(i_m|A_{\tau k},\theta^*)\}\, x_{\tau i_m} - \lambda_t\theta^*$$

$$= \sum_{\tau=1}^{t}\sum_{k=1}^{O_\tau}\sum_{i_m\in A_{\tau k}} \epsilon_{\tau km}x_{\tau i_m} - \lambda_t\theta^*$$

$$=: M_t - \lambda_t\theta^*.$$

Thus, we have:

$$||g_t(\hat{\theta}_t) - g_t(\theta^*)||_{H_t^{-1}} \leq ||M_t||_{H_t^{-1}} + \lambda_t||\theta^*||_{H_t^{-1}}. \tag{24}$$

Now, we need to bound $\lambda_t||\theta^*||_{H_t^{-1}}$. We have

$$||\theta^*||_{H_t^{-1}}^2 \leq \frac{||\theta^*||_2^2}{\lambda_{\min}(H_t)} \leq \frac{||\theta^*||_2^2}{\lambda_{\min}(\lambda_t I_d)} \leq \frac{||\theta^*||_2^2}{\lambda_t}.$$

By Assumption 4.1 that $||\theta^*||_2^2 \leq 1$, $\lambda_t ||\theta^*||_{H_t^{-1}} \leq \sqrt{\lambda_t}$. We can rewritten Equation (24) as follows:

$$||g_t(\hat{\theta}_t) - g_t(\theta^*)||_{H_t^{-1}} \leq ||M_t||_{H_t^{-1}} + \sqrt{\lambda_t}. \tag{25}$$

Since the reward at any round and for any cascade is constrained to be no more than 1, given the filtration $\mathcal{F}_t$, $\epsilon_{\tau km}$ which is upper bounded by 1 behaves as a martingale difference. To apply Theorem C.2, we calculate $\forall \tau \geq 1$:

$$\mathbb{E}\left[\epsilon_{\tau km}^2 \mid \mathcal{F}_t\right] = \mathbb{E}\left[\{y_{\tau km} - p_\tau(i_m|A_{\tau k}, \theta^*)\}^2 \mid \mathcal{F}_t\right]$$
$$= \mathbb{V}\left[y_{\tau km} \mid \mathcal{F}_t\right] = p_\tau(i_m|A_{\tau k}, \theta^*)(1 - p_\tau(i_m|A_{\tau k}, \theta^*))$$
$$=: \dot{p}_\tau(i_m|A_{\tau k}, \theta^*).$$

Then, setting $\mathbb{H}_t$ as $H_t$ and $\mathbb{M}_t$ as $M_t$, we obtain:

$$1 - \delta \leq \mathbb{P}\left(\forall t \geq 1, ||M_t||_{H_t^{-1}} \leq \frac{\sqrt{\lambda_t}}{2} + \frac{2}{\sqrt{\lambda_t}} \log\left(\frac{\det(H_t)^{\frac{1}{2}} \lambda_t^{-\frac{d}{2}}}{\delta}\right) + \frac{2}{\sqrt{\lambda_t}} d \log 2\right). \tag{26}$$

Now, we need to bound $\det(H_t)$.

$$\det(H_t) = \det\left(\sum_{\tau=1}^{t} \sum_{k\in[O_\tau]} \sum_{i\in A_{\tau k}} \dot{p}_\tau(i|A_{\tau k}, \theta^*) x_{\tau i} x_{\tau i}^\top + \lambda_t I_d\right)$$
$$\leq \det\left(\sum_{\tau=1}^{t-1} \sum_{k\in[O_\tau]} \sum_{i\in A_{\tau k}} x_{\tau i} x_{\tau i}^\top + \lambda_t I_d\right)$$
$$= \det(V_t) \leq \left(\lambda_t + \frac{tKM}{d}\right)^d.$$

The first inequality is from the fact that $\dot{p}_\tau(i|A_{\tau k}, \theta^*) \leq 1$ and the last inequality is obtained by using Lemma B.9. We substitute the above results into Eq.(26). This simplify Eq.(26) as:

$$1 - \delta \leq \mathbb{P}\left(\forall t \geq 1, ||M_t||_{H_t^{-1}} \leq \frac{\sqrt{\lambda_t}}{2} + \frac{2}{\sqrt{\lambda_t}} \log\left(\frac{(\lambda_t + tKM/d)^{\frac{d}{2}} \lambda_t^{-\frac{d}{2}}}{\delta}\right) + \frac{2}{\sqrt{\lambda_t}} d \log 2\right).$$

Thus, the following holds with probability at least $1 - \delta$ by combining the above result and Equation (25):

$$||g_t(\hat{\theta}_t) - g_t(\theta^*)||_{H_t^{-1}} \leq \frac{3\sqrt{\lambda_t}}{2} + \frac{2}{\sqrt{\lambda_t}} \log\left(\frac{(\lambda_t + tKM/d)^{\frac{d}{2}} \lambda_t^{-\frac{d}{2}}}{\delta}\right) + \frac{2}{\sqrt{\lambda_t}} d \log 2.$$

$\square$

**Lemma C.4** (Lemma 12 in [3]). *For an assortment $A_{tk}$ and $\theta_1, \theta_2 \in \mathbb{R}_d$, the following holds:*

$$\sum_{i\in A_{tk}} \int_{v=0}^{1} \dot{p}_t(i|A_{tk}, v\theta_2 + (1-v)\theta_1) \cdot dv \geq \sum_{i\in A_{tk}} \dot{p}_t(i|A_{tk}, \theta_1)(1 + |x_{ti}^\top \theta_1 - x_{ti}^\top \theta_2|)^{-1}.$$

**Lemma C.5.** *For any $\theta_1, \theta_2 \in \mathbb{R}^d$, the followings hold:*

$$\mathbb{G}_t(\theta_1, \theta_2) \succeq \frac{1}{3} H_t(\theta_1)$$

$$\mathbb{G}_t(\theta_1, \theta_2) \succeq \frac{1}{3} H_t(\theta_2)$$

*where $\mathbb{G}_t(\theta_1, \theta_2) = \sum_{\tau=1}^{t} \sum_{k=1}^{O_\tau} \sum_{i\in A_{\tau k}} \left[\int_{v=0}^{1} \dot{p}_\tau(i|A_{\tau k}, v\theta_2 + (1-v)\theta_1)dv\right] x_{\tau i} x_{\tau i}^\top + \lambda_t I_d$ and $I_d$ is d-dimensional identitiy matrix.*

*Proof.*

$$\sum_{i \in A_{tk}} \int_{v=0}^{1} \dot{p}_t(i|A_{tk}, v\theta_2 + (1-v)\theta_1) \cdot dv \geq \sum_{i \in A_{tk}} \dot{p}_t(i|A_{tk}, \theta_1)(1 + |x_{ti}^\top \theta_1 - x_{ti}^\top \theta_2|)^{-1}$$

$$\geq \sum_{i \in A_{tk}} \dot{p}_t(i|A_{tk}, \theta_1)(1 + ||x_{ti}||_2||\theta_1 - \theta_2||_2)^{-1}$$

$$\geq \sum_{i \in A_{tk}} \frac{\dot{p}_t(i|A_{tk}, \theta_1)}{3}.$$

The first inequality is from Lemma C.4, the second inequality is obtained by applying Cachy-Schwarz inequality and the last inequality is from our Assumption 4.1. The definition of $\mathbb{G}_t(\theta_1, \theta_2)$ is as follows:

$$\mathbb{G}_t(\theta_1, \theta_2) = \sum_{\tau=1}^{t} \sum_{k=1}^{O_\tau} \sum_{i \in A_{\tau k}} \left[ \int_{v=0}^{1} \dot{p}_\tau(i|A_{\tau k}, v\theta_2 + (1-v)\theta_1)dv \right] x_{\tau i} x_{\tau i}^\top + \lambda_t I_d$$

$$\succeq \frac{1}{3} \sum_{\tau=1}^{t} \sum_{k \in [O_\tau]} \sum_{i \in A_{\tau k}} \dot{p}_t(i|A_{tk}, \theta_1) x_{\tau i} x_{\tau i}^\top + \lambda_t I_d$$

$$= \frac{1}{3} \left\{ \sum_{\tau=1}^{t} \sum_{k \in [O_\tau]} \sum_{i \in A_{\tau k}} \dot{p}_t(i|A_{tk}, \theta_1) x_{\tau i} x_{\tau i}^\top + (1+2)\lambda_t I_d \right\}$$

$$\succeq \frac{1}{3} \left\{ \sum_{\tau=1}^{t} \sum_{k \in [O_\tau]} \sum_{i \in A_{\tau k}} \dot{p}_t(i|A_{tk}, \theta_1) x_{\tau i} x_{\tau i}^\top + \lambda_t I_d \right\}$$

$$= \frac{1}{3} H_t(\theta_1).$$

For deriving the second relationship in Lemma C.5, given that $\theta_1$ and $\theta_2$ possess symmetrical roles in $\int_{v=0}^{1} \dot{p}_\tau(i|A_{\tau k}, v\theta_2 + (1-v)\theta_1)dv$, we substitute two parameters and can get the result. $\square$

**Lemma C.6** (Restatement of Lemma 5.1). *Suppose $\theta^* \in B_t(\delta)$. Then, for any $\theta \in B_t(\delta)$, we have*

$$||\theta - \theta^*||_{H_t} \leq 6\gamma_t(\delta). \tag{27}$$

*where the confidence radius* $\gamma_t(\delta) := \frac{3\sqrt{\lambda_t}}{2} + \frac{2}{\sqrt{\lambda_t}} \log\left( \frac{(\lambda_t + tKM/d)^{d/2} \lambda_t^{-d/2}}{\delta} \right) + \frac{2d}{\sqrt{\lambda_t}} \log 2.$

*Proof.*

$$||\theta - \theta^*||_{H_t} \leq \sqrt{3}||\theta - \theta^*||_{\mathbb{G}_t(\theta, \theta^*)}$$

$$= \sqrt{3}||g_t(\theta^*) - g_t(\theta)||_{\mathbb{G}_t^{-1}(\theta, \theta^*)}$$

$$\leq \sqrt{3} \left\{ ||g_t(\theta^*) - g_t(\hat{\theta}_t)||_{\mathbb{G}_t^{-1}(\theta, \theta^*)} + ||g_t(\hat{\theta}_t) - g_t(\theta)||_{\mathbb{G}_t^{-1}(\theta, \theta^*)} \right\}$$

$$\leq \sqrt{3} \left\{ \sqrt{3}||g_t(\theta^*) - g_t(\hat{\theta}_t)||_{H_t^{-1}} + \sqrt{3}||g_t(\hat{\theta}_t) - g_t(\theta)||_{H_t^{-1}} \right\}$$

$$= 3 \left\{ ||g_t(\theta^*) - g_t(\hat{\theta}_t)||_{H_t^{-1}} + ||g_t(\hat{\theta}_t) - g_t(\theta)||_{H_t^{-1}} \right\}$$

$$= 6\gamma_t(\delta).$$

The first and third inequality is from Lemma C.5. The first equality is from Lemma C.1. From triangle inequality, we obtain the second inequality. Since $\theta \in B_t(\delta)$, the last equality holds. $\square$

The above lemma shows the upper bound on the difference between an arbitrary $\theta$ and $\theta^*$, and is pivotal for determining the overall regret bound.

**Lemma C.7.** *For the cascade $S_t$ chosen by* UCB-CCA+ *and any $\theta_t \in B_t(\delta)$, the following holds with probability at least $1 - \delta$:*

$$f(S_t, \theta_t) - f(S_t, \theta^*) \leq 36\,\gamma_t(\delta) \max_{\substack{A_{tk} \in S_t \\ i \in A_{tk}}} \sqrt{\dot{p}_t(i|A_{tk}, \theta^*)} \|x_{ti}\|_{H_{t-1}^{-1}} + \frac{216}{\kappa}\gamma_t^2(\delta) \max_{\substack{A_{tk} \in S_t \\ i \in A_{tk}}} \|x_{ti}\|_{V_{t-1}^{-1}}^2$$

*Proof.* We begin by performing a second Taylor expansion of the expected reward function:

$$
\begin{aligned}
&f(S_t, \theta_t) - f(S_t, \theta^*) \\
&= \nabla_\theta f(S_t, \theta^*)(\theta_t - \theta^*) + (\theta_t - \theta^*)^\top \left[ \int_{v=0}^1 (1-v)\nabla_\theta^2 f(S_t, \bar{\theta}))\, dv \right](\theta_t - \theta^*) \\
&\leq \sum_{A_{tk} \in S_t} \sum_{i \in A_{tk}} \left\{ \prod_{\substack{A_{t\dot{k}} \in \\ S_t \backslash \{A_{tk}\}}} p_t(i_0|A_{t\dot{k}}, \theta^*) \right\} \dot{p}_t(i|A_{tk}, \theta^*) x_{ti}^\top(\theta_t - \theta^*) \\
&\quad + \int_{v=0}^1 (1-v) \sum_{A_{tk} \in S_t} \sum_{i \in A_{tk}} \left\{ \prod_{\substack{A_{t\dot{k}} \in \\ S_t \backslash \{A_{tk}\}}} p_t(i_0|A_{t\dot{k}}, \bar{\theta}) \right\} \ddot{p}_t(i|A_{tk}, \bar{\theta}) dv(x_{ti}^\top(\theta_t - \theta^*))^2 \qquad (28)
\end{aligned}
$$

where $\bar{\theta} := \theta^* + v(\theta_t - \theta^*)$.

Then, we bound $\sum_{A_{tk} \in S_t} \sum_{i \in A_{tk}} \left\{ \prod_{\substack{A_{t\dot{k}} \in \\ S_t \backslash \{A_{tk}\}}} p_t(i_0|A_{t\dot{k}}, \theta^*) \right\}$ as follows:

$$
\begin{aligned}
\sum_{A_{tk} \in S_t} \sum_{i \in A_{tk}} \left\{ \prod_{\substack{A_{t\dot{k}} \in \\ S_t \backslash \{A_{tk}\}}} p_t(i_0|A_{t\dot{k}}, \theta^*) \right\} &\leq \sum_{A_{tk} \in S_t} \sum_{i \in A_{tk}} \left\{ \frac{1}{1 + M\exp(-1)} \right\}^{K-1} \\
&= KM \left\{ \frac{1}{1 + M\exp(-1)} \right\}^{K-1} \\
&= K \left\{ \frac{1}{1 + M\exp(-1)} \right\}^{K-2} \cdot M \left\{ \frac{1}{1 + M\exp(-1)} \right\} \\
&\leq K \left\{ \frac{1}{1 + \exp(-1)} \right\}^{K-2} \cdot \exp(1) \\
&\leq \frac{(1 + \exp(1))^2}{\exp(1)^3(\log(1 + \exp(1)) - 1)} \cdot \exp(1) \\
&< \frac{(1 + \exp(1))^2}{\exp(1)^2(\log(1 + \exp(1)) - 1)} < 6. \qquad (29)
\end{aligned}
$$

The first inequality is due to Assumption 4.1 and the definition of $p_t(i_0|A_{t\dot{k}}, \theta^*)$. Since $K \left\{ \frac{1}{1+\exp(-1)} \right\}^{K-2}$ is maximized as $\frac{(1+\exp(1))^2}{\exp(1)^3(\log(1+\exp(1))-1)}$ at $K = \frac{1}{\log(1+\exp)-1} \approx 3.19$, we

can derive the third inequality. We apply Equation (29) to Equation (28) as follows:

$$f\left(S_t, \theta_t\right) - f\left(S_t, \theta^*\right) \le 6 \max_{\substack{A_{tk} \in S_t \\ i \in A_{tk}}} \dot{p}_t(i|A_{tk}, \theta^*) x_{ti}^\top (\theta_t - \theta^*)$$

$$+ 6 \max_{\substack{A_{tk} \in S_t \\ i \in A_{tk}}} \left[ \int_{v=0}^1 (1-v) \ddot{p}_t(i|A_{tk}, \bar{\theta}) dv \right] (x_{ti}^\top (\theta_t - \theta^*))^2$$

$$\le 6 \max_{\substack{A_{tk} \in S_t \\ i \in A_{tk}}} \dot{p}_t(i|A_{tk}, \theta^*) ||x_{ti}||_{H_{t-1}^{-1}} ||\theta_t - \theta^*||_{H_{t-1}}$$

$$+ 6 \max_{\substack{A_{tk} \in S_t \\ i \in A_{tk}}} \left[ \int_{v=0}^1 (1-v) \ddot{p}_t(i|A_{tk}, \bar{\theta}) dv \right] ||x_{ti}||_{H_{t-1}^{-1}}^2 ||\theta_t - \theta^*||_{H_{t-1}}^2$$

$$\le 36 \, \gamma_t(\delta) \max_{\substack{A_{tk} \in S_t \\ i \in A_{tk}}} \dot{p}_t(i|A_{tk}, \theta^*) ||x_{ti}||_{H_{t-1}^{-1}} + 216 \, \gamma_t^2(\delta) \max_{\substack{A_{tk} \in S_t \\ i \in A_{tk}}} ||x_{ti}||_{H_{t-1}^{-1}}^2$$

$$\le 36 \, \gamma_t(\delta) \max_{\substack{A_{tk} \in S_t \\ i \in A_{tk}}} \sqrt{\dot{p}_t(i|A_{tk}, \theta^*)} ||x_{ti}||_{H_{t-1}^{-1}} + \frac{216}{\kappa} \gamma_t^2(\delta) \max_{\substack{A_{tk} \in S_t \\ i \in A_{tk}}} ||x_{ti}||_{V_{t-1}^{-1}}^2$$

$\square$

**Lemma C.8** (Elliptical potential with local information).

$$\sum_{t=1}^T \max_{\substack{k \in [O_t] \\ i \in A_{tk}}} ||\sqrt{\dot{p}_t(i|A_{tk}, \theta^*)} x_{ti}||_{H_{t-1}^{-1}}^2 \le 2d \log\left(1 + \frac{KMT}{d\lambda}\right)$$

*Proof.* We begin by demonstrating that the weighted $\ell_2$ norm of the feature vector combined with the local information inside the above max operator is upper bounded by 1. For convenience, $\sqrt{\dot{p}_t(i|A_{tk}, \theta^*)} x_{ti}$ is denoted by $\tilde{x}_{ti}$. So, we rewrite $H_t := \sum_{\tau=1}^t \sum_{k \in [O_\tau]} \sum_{i \in A_{\tau k}} \tilde{x}_{\tau i} \tilde{x}_{\tau i}^\top + \lambda_t I_d$. Let $\lambda_{\min}(H_t)$ be the minimum eigenvalue of $H_t$. Since $\lambda_t \ge 1$ and $||\tilde{x}_{ti}||_{H_{t-1}^{-1}}^2 \le \frac{||\tilde{x}_{ti}||_2^2}{\lambda_{\min}(H_{t-1})} \le \frac{1}{\lambda_t}$, we have:

$$\max_{k \in [O_t]} \max_{i \in A_{tk}} ||\tilde{x}_{ti}||_{H_{t-1}^{-1}}^2 \le 1.$$

Using the fact that $z \le 2\ln(1+z)$ for any $z \in [0,1]$, we have

$$\sum_{\tau=1}^t \max_{k \in [O_\tau]} \max_{i \in A_{\tau k}} ||\tilde{x}_{\tau i}||_{H_{\tau-1}^{-1}}^2 \le 2 \sum_{\tau=1}^t \ln\left(1 + \max_{k \in [O_\tau]} \max_{i \in A_{\tau k}} ||\tilde{x}_{\tau i}||_{H_{\tau-1}^{-1}}^2\right)$$

$$= 2\ln \prod_{\tau=1}^t \left(1 + \max_{k \in [O_\tau]} \max_{i \in A_{\tau k}} ||\tilde{x}_{\tau i}||_{H_{\tau-1}^{-1}}^2\right). \tag{30}$$

Now we upper bound $\prod_{\tau=1}^{t} \left( 1 + \max_{k \in [O_\tau]} \max_{i \in A_{\tau k}} ||\tilde{x}_{\tau i}||^2_{H^{-1}_{\tau-1}} \right)$ from $\det(H_t)$.

$$
\begin{aligned}
\det(H_t) &= \det \left( H_{t-1} + \sum_{k=1}^{O_t} \sum_{i \in A_{tk}} \tilde{x}_{ti} \tilde{x}_{ti}^\top \right) \\
&= \det(H_{t-1}) \det \left( I + H_{t-1}^{-1/2} \sum_{k=1}^{O_t} \sum_{i \in A_{tk}} \tilde{x}_{ti} \tilde{x}_{ti}^\top H_{t-1}^{-1/2} \right) \\
&= \det(H_{t-1}) \det \left( I + \sum_{k=1}^{O_t} \sum_{i \in A_{tk}} \left( H_{t-1}^{-1/2} \tilde{x}_{ti} \right) \left( H_{t-1}^{-1/2} \tilde{x}_{ti} \right)^\top \right) \\
&\geq \det(H_{t-1}) \left( 1 + \sum_{k=1}^{O_t} \sum_{i \in A_{tk}} ||\tilde{x}_{ti}||^2_{H^{-1}_{t-1}} \right) \\
&\geq \det(\lambda I) \prod_{\tau=1}^{t} \left( 1 + \sum_{k=1}^{O_\tau} \sum_{i \in A_{\tau k}} ||\tilde{x}_{\tau i}||^2_{H^{-1}_{\tau-1}} \right) \\
&\geq \det(\lambda I) \prod_{\tau=1}^{t} \left( 1 + \max_{k \in [O_\tau]} \max_{i \in A_{\tau k}} ||\tilde{x}_{\tau i}||^2_{H^{-1}_{\tau-1}} \right).
\end{aligned}
$$

The second equality above is from the fact that $V + U = V^{1/2}(I + V^{-1/2}UV^{-1/2})V^{1/2}$ for a symmetric positive definite matrix $V$. The first inequality above can be obtained by applying Lemma B.7. Applying the first inequality repeatedly, we can get the second inequality above. Thus, we have

$$
\prod_{\tau=1}^{t} \left( 1 + \max_{k \in [O_\tau]} \max_{i \in A_{\tau k}} ||\tilde{x}_{\tau i}||^2_{H^{-1}_{\tau-1}} \right) \leq \frac{\det(H_t)}{\det(\lambda I)}. \tag{31}
$$

Then applying Eq.(31) to Eq.(30), we complete the proof as follows:

$$
\sum_{\tau=1}^{t} \max_{k \in [O_\tau]} \max_{i \in A_{\tau k}} ||\tilde{x}_{\tau i}||^2_{H^{-1}_{\tau-1}} \leq 2 \ln \frac{\det(H_t)}{\det(\lambda I)} \leq 2 \ln \frac{\det(H_t)}{\lambda^d} \leq 2 \ln \frac{\det(V_t)}{\lambda^d} \leq 2d \log \left( 1 + \frac{KMT}{d\lambda} \right)
$$

where the last inequality is from Lemma B.9. □

*Proof of Theorem 5.2.*

$$\mathcal{R}^\alpha(T) = \mathbb{E}\left[\sum_{t=1}^T \mathcal{R}^\alpha(t, S_t)\right] \leq \mathbb{E}\left[\sum_{t=1}^T f(S_t, \theta_t) - f(S_t, \theta^*)\right]$$

$$\leq \mathbb{E}\left[\sum_{t=1}^T 36\,\gamma_t(\delta) \max_{\substack{A_{tk} \in S_t \\ i \in A_{tk}}} \sqrt{\dot{p}_t(i|A_{tk}, \theta^*)}||x_{ti}||_{H_{t-1}^{-1}} + \sum_{t=1}^T \frac{216}{\kappa}\gamma_t^2(\delta) \max_{\substack{A_{tk} \in S_t \\ i \in A_{tk}}} ||x_{ti}||_{V_{t-1}^{-1}}^2\right] \quad (32)$$

$$\leq \mathbb{E}\left[36\,\gamma_T(\delta)\sum_{t=1}^T \max_{\substack{A_{tk} \in S_t \\ i \in A_{tk}}} ||\sqrt{\dot{p}_t(i|A_{tk}, \theta^*)}x_{ti}||_{H_{t-1}^{-1}} + \frac{216}{\kappa}\gamma_T^2(\delta)\sum_{t=1}^T \max_{\substack{A_{tk} \in S_t \\ i \in A_{tk}}} ||x_{ti}||_{V_t^{-1}}^2\right]$$

$$\leq \mathbb{E}\left[36\,\gamma_T(\delta)\sum_{t=1}^T \max_{\substack{k \in [O_t] \\ i \in A_{tk}}} ||\sqrt{\dot{p}_t(i|A_{tk}, \theta^*)}x_{ti}||_{H_{t-1}^{-1}}\right] \quad (33)$$

$$+ \mathbb{E}\left[\frac{216}{\kappa p_t(i_0|A_{t1}, \theta^*)}\gamma_T^2(\delta)\sum_{t=1}^T \max_{\substack{k \in [O_t] \\ i \in A_{tk}}} ||x_{ti}||_{V_{t-1}^{-1}}^2\right] \quad (34)$$

$$\leq \mathbb{E}\left[36\,\gamma_T(\delta)\sqrt{T \cdot \sum_{t=1}^T \max_{\substack{k \in [O_t] \\ i \in A_{tk}}} ||\sqrt{\dot{p}_t(i|A_{tk}, \theta^*)}x_{ti}||_{H_{t-1}^{-1}}^2}\right] \quad (35)$$

$$+ \mathbb{E}\left[\frac{216(1 + Me)}{\kappa}\gamma_T^2(\delta)\sum_{t=1}^T \max_{\substack{k \in [O_t] \\ i \in A_{tk}}} ||x_{ti}||_{V_t^{-1}}^2\right] \quad (36)$$

$$\leq 36\gamma_T(\delta)\sqrt{T \cdot 2d\log\left(1 + \frac{KMT}{d\lambda_t}\right)} + \frac{216(1 + Me)}{\kappa}\gamma_T^2(\delta)\left(d\log\left(1 + \frac{KMT}{d\lambda_t}\right)\right) \quad (37)$$

The inequality (32) is from Lemma C.7. The inequality (34) can be obtained by *doubly optimistic exposure swapping* (see Section 5.1.2 and max operation). The Inequality (36) is from Cauchy-Schwarz inequality. The last inequality is from Lemma C.8, Lemma B.8 and Lemma B.9. $\square$

# D  Convex Relaxation

In `UCB-CCA+`'s optimization step (see line 4 in Algorithm 2), it is particularly challenging to solve due to the confidence set $B_t(\delta)$ in Eq.(6) being a non-convex set. [2; 3] address this problem by designing a convex relaxation for the set $B_t(\delta)$ in simple logistic bandits and MNL bandits, respectively, and we extend this idea to our model. The following confidence set is a convex relation set for $B_t(\delta)$:

$$E_t(\delta) := \left\{\theta \in \mathbb{R}^d : \mathcal{L}_t(\theta) - \mathcal{L}_t(\hat{\theta}_t) \leq \zeta_t^2(\delta)\right\}$$

where $\zeta_t(\delta) := \gamma_t(\delta) + \frac{\gamma_t^2(\delta)}{\sqrt{\lambda_t}}$ and $\gamma_t(\delta) := \frac{3\sqrt{\lambda_t}}{2} + \frac{2}{\sqrt{\lambda_t}}\log\left(\frac{(\lambda_t + KMt/d)^{d/2}\lambda_t^{-d/2}}{\delta}\right) + \frac{2d}{\sqrt{\lambda_t}}\log 2$.

We exploit $E_t(\delta)$ instead of the original confidence set $B_t(\delta)$ for numerical experiments in Section 7. In this section, we justify our strategy of relaxing the confidence set based on the following lemmas.

**Lemma D.1.** $\forall t \geq 1, E_t(\delta) \supseteq B_t(\delta)$, *therefore* $\mathbb{P}(\forall t \geq 1, \theta^* \in E_t(\delta)) \geq 1 - \delta$.

*Proof.* We start by performing the second-order Taylor expansion of the log-likelihood function with respect to $\hat{\theta}_t$ as follows:

$$
\mathcal{L}_t(\theta) - \mathcal{L}_t(\hat{\theta}_t)
$$

$$
= \nabla\mathcal{L}_t(\hat{\theta}_t)^\top(\theta - \hat{\theta}_t) + (\theta - \hat{\theta}_t)^\top \left(\int_{v=0}^1 (1-v)\nabla^2\mathcal{L}_t(\hat{\theta}_t + v(\theta - \hat{\theta}_t))dv\right)(\theta - \hat{\theta}_t) \quad (38)
$$

$$
= (\theta - \hat{\theta}_t)^\top \left(\int_{v=0}^1 (1-v)\nabla^2\mathcal{L}_t(\hat{\theta}_t + v(\theta - \hat{\theta}_t))dv\right)(\theta - \hat{\theta}_t)
$$

$$
= (\theta - \hat{\theta}_t)^\top \left(\int_{v=0}^1 (1-v)H_t(\hat{\theta}_t + v(\theta - \hat{\theta}_t))dv\right)(\theta - \hat{\theta}_t) \quad (39)
$$

$$
\leq (\theta - \hat{\theta}_t)^\top \mathbb{G}_t(\hat{\theta}_t - \theta)(\theta - \hat{\theta}_t) \quad (40)
$$

$$
= \|\theta - \hat{\theta}_t\|_{\mathbb{G}_t(\hat{\theta}_t,\theta)}^2
$$

$$
= \|g_t(\theta) - g_t(\hat{\theta}_t)\|_{\mathbb{G}_t^{-1}(\hat{\theta}_t,\theta)}^2 \quad (41)
$$

$$
= \|g_t(\theta) - g_t(\hat{\theta}_t)\|_{\mathbb{G}_t^{-1}(\theta,\hat{\theta}_t)}^2. \quad (42)
$$

Equation (38) from the fact that $\nabla\mathcal{L}_t(\hat{\theta}_t) = 0$. Equation (39) is due to $\nabla^2\mathcal{L}_t(\theta) = H_t(\theta)$. Equation (40) is derived through the following process:

$$
\int_{v=0}^1 (1-v)H_t(\hat{\theta}_t + v(\theta - \hat{\theta}_t))dv
$$

$$
= \sum_{\tau=1}^t \sum_{k=1}^{O_\tau} \sum_{i\in A_{\tau k}} \left(\int_{v=0}^1 (1-v)\dot{p}_\tau(i|A_{\tau k},\hat{\theta}_t + v(\theta - \hat{\theta}_t)dv\right) x_{\tau i}x_{\tau i}^\top + \lambda_\tau I_d
$$

$$
\preceq \sum_{\tau=1}^t \sum_{k=1}^{O_\tau} \sum_{i\in A_{\tau k}} \left(\int_{v=0}^1 \dot{p}_\tau(i|A_{\tau k},\hat{\theta}_t + v(\theta - \hat{\theta}_t)dv\right) x_{\tau i}x_{\tau i}^\top + \lambda_\tau I_d \quad (43)
$$

$$
= \mathbb{G}_t(\hat{\theta}_t - \theta) \quad (44)
$$

Equation (43) holds because $p_t(i|A,\theta,)$ is a strictly increasing function with respect to $\theta$ ( $\dot{p}_t(i|A,\theta,)$). And Equation (44) is from the definition of $\mathbb{G}_t(\theta_1 - \theta_2)$ as defined in Lemma C.5. Equation (41) is obtained by applying Lemma C.1 Equation (42) is from the fact that $\mathbb{G}_t^{-1}(\hat{\theta}_t,\theta) = \mathbb{G}_t^{-1}(\theta,\hat{\theta}_t)$. Now, we apply Lemma D.2 to Equation (42), then we have:

$$
\mathcal{L}_t(\theta) - \mathcal{L}_t(\hat{\theta}_t) \leq \left\{\gamma_t(\delta) + \frac{\gamma_t^2(\delta)}{\sqrt{\lambda_t}}\right\}^2 = \zeta_t^2(\delta)
$$

for any $\theta \in B_t(\delta)$. This shows that if $\theta \in B_t(\delta)$, then $\theta \in E_t(\delta)$ and thus $B_t(\delta) \subset E_t(\delta)$. $\quad\square$

**Lemma D.2.** *Let $\delta \in (0,1]$. For all $\theta \in B_t(\delta)$:*

$$
\|g_t(\theta) - g_t(\hat{\theta}_t)\|_{\mathbb{G}_t^{-1}(\theta,\hat{\theta}_t)} \leq \gamma_t(\delta) + \frac{\gamma_t^2(\delta)}{\sqrt{\lambda_t}} = \zeta_t(\delta)
$$

*Proof.*

$$\mathbb{G}_t(\theta, \hat{\theta}_t) = \sum_{\tau=1}^{t1} \sum_{k \in [O_\tau]} \sum_{i \in A_{\tau k}} \alpha_i(A_{tk}, \hat{\theta}_t, \theta) x_{\tau i} x_{\tau i}^\top + \lambda_t I_d$$

$$\geq \sum_{\tau=1}^{t1} \sum_{k \in [O_\tau]} \sum_{i \in A_{\tau k}} \dot{p}_\tau(i|A_{\tau k}, \theta)(1 + |x_{\tau i}^\top \theta - x_{\tau i}^\top \hat{\theta}_t|)^{-1} x_{\tau i} x_{\tau i}^\top + \lambda_t I_d$$

$$\geq \sum_{\tau=1}^{t1} \sum_{k \in [O_\tau]} \sum_{i \in A_{\tau k}} \dot{p}_\tau(i|A_{\tau k}, \theta)(1 + ||x_{\tau i}||_{\mathbb{G}_t^{-1}(\theta, \hat{\theta}_t)} ||\theta - \hat{\theta}_t||_{\mathbb{G}_t(\theta, \hat{\theta}_t)})^{-1} x_{\tau i} x_{\tau i}^\top + \lambda_t I_d$$

$$\geq (1 + \lambda_t^{-1/2} ||\theta - \hat{\theta}_t||_{\mathbb{G}_t(\theta, \hat{\theta}_t)})^{-1} \left( \sum_{\tau=1}^{t} \sum_{k \in [O_\tau]} \sum_{i \in A_{\tau k}} \dot{p}_\tau(i|A_{\tau k}, \theta) x_{\tau i} x_{\tau i}^\top + \lambda_t I_d \right)$$

$$= (1 + \lambda_t^{-1/2} ||\theta - \hat{\theta}_t||_{\mathbb{G}_t(\theta, \hat{\theta}_t)})^{-1} H_t(\theta)$$

$$= (1 + \lambda_t^{-1/2} ||g_t(\theta) - g_t(\hat{\theta}_t)||_{\mathbb{G}_t^{-1}(\theta, \hat{\theta}_t)})^{-1} H_t(\theta).$$

The first inequality is from Lemma C.4 and the second inequality is obtained by applying Cauchy-Schwarz inequality.

This inequality gives:

$$||g_t(\theta) - g_t(\hat{\theta}_t)||_{\mathbb{G}_t^{-1}(\theta, \hat{\theta}_t)}^2 \leq \left\{ 1 + \lambda_t^{-1/2} ||g_t(\theta) - g_t(\hat{\theta}_t)||_{\mathbb{G}_t^{-1}(\theta, \hat{\theta}_t)} \right\} ||g_t(\theta) - g_t(\hat{\theta}_t)||_{H_t(\theta)^{-1}}^2$$

$$\leq \lambda_t^{-1/2} \gamma_t^2(\delta) ||g_t(\theta) - g_t(\hat{\theta}_t)||_{\mathbb{G}_t^{-1}(\theta, \hat{\theta}_t)} + \gamma_t^2(\delta).$$

Resolving this polynomial inequality with respect to $||g_t(\theta) - g_t(\hat{\theta}_t)||_{\mathbb{G}_t^{-1}(\theta, \hat{\theta}_t)}$ by using the fact that $x^2 \leq bx + c \Rightarrow x \leq b + \sqrt{c}$ where $x \in \mathbb{R}$ and $b, c \in \mathbb{R}+$ (See Proposition 7 in Abeille et al. [2])., then we get the result. $\qquad \square$

**Lemma D.3.** *Suppose $\theta^* \in B_t(\delta)$, the following holds for all $\theta \in E_t(\delta)$:*
$$||\theta - \theta^*||_{H_t} \leq 4\gamma_t(\delta) + 2\sqrt{2}\zeta_t(\delta).$$

*Proof.* We start by performing the second-order Taylor expansion of the log-likelihood function with respect to $\theta^*$ as follows:

$$\mathcal{L}_t(\theta) - \mathcal{L}_t(\theta^*)$$

$$= \nabla \mathcal{L}_t(\theta^*)^\top (\theta - \theta^*) + (\theta - \theta^*)^\top \left( \int_{v=0}^{1} (1 - v) \nabla^2 \mathcal{L}_t(\theta^* + v(\theta - \theta^*)) dv \right) (\theta - \theta^*)$$

$$= \nabla \mathcal{L}_t(\theta^*)^\top (\theta - \theta^*) + ||\theta - \theta^*||_{\tilde{\mathbb{G}}_t(\theta^*, \theta)}^2$$

$$\geq \nabla \mathcal{L}_t(\theta^*)^\top (\theta - \theta^*) + \frac{1}{4} ||\theta - \theta^*||_{H_t}^2$$

where $\tilde{\mathbb{G}}_t(\theta^*, \theta) = (\theta - \theta^*)^\top \left( \int_{v=0}^{1} (1 - v) H_t(\theta^* + v(\theta - \theta^*)) dv \right) (\theta - \theta^*)$. The last inequality is from applying Lemma 8 in Abeille et al. [2]. Thus, we have:

$$||\theta - \theta^*||_{H_t}^2 \leq 4|\mathcal{L}_t(\theta) - \mathcal{L}_t(\theta^*)| + 4|\nabla \mathcal{L}_t(\theta^*)^\top (\theta - \theta^*)|$$

$$\leq 8\zeta_t(\delta)^2 + 4|\nabla \mathcal{L}_t(\theta^*)^\top (\theta - \theta^*)|$$

$$\leq 8\zeta_t(\delta)^2 + 4||\nabla \mathcal{L}_t(\theta^*)||_{H_t^{-1}} ||\theta - \theta^*||_{H_t}$$

$$\leq 8\zeta_t(\delta)^2 + 4\gamma_t(\delta) ||\theta - \theta^*||_{H_t}. \tag{45}$$

The above second inequality is due to $\theta, \theta^* \in E_t(\delta)$. The third inequality is by applying Cachy-Schwarz inequality. And Equation (45) holds from the following inequality:

$$||\nabla \mathcal{L}_t(\theta^*)||_{H_t^{-1}} = ||g_t(\theta^*) - \sum_{\tau=1}^{t} \sum_{k=1}^{O_\tau} \sum_{i_m \in A_{\tau k}} y_{\tau k m} x_{\tau i_m}||_{H_t^{-1}} \tag{46}$$

$$= ||g_t(\theta^*) - g_t(\hat{\theta}_t)||_{H_t^{-1}} \leq \gamma_t(\delta). \tag{47}$$

Equation (46) is from the definition of $\nabla \mathcal{L}_t(\theta^*)$ and $g_t(\theta^*)$. Equation (47) is from Equation (22). In conclusion, Equation (45) is a polynomial inequality in terms of $||\theta - \theta^*||_{H_t}$. Solving it yields the following result:

$$||\theta - \theta^*||_{H_t} \leq 4\gamma_t(\delta) + 2\sqrt{2}\zeta_t(\delta).$$

since $x^2 \leq bx + c \Rightarrow x \leq b + \sqrt{c}$ where $x \in \mathbb{R}$ and $b, c \in \mathbb{R}+$ (See Proposition 7 in Abeille et al. [2]).

Lastly, we set the penalty parameter $\lambda_t = \mathcal{O}(d \log(tKM))$, then we get the followings:

$$\gamma_t(\delta) = \mathcal{O}(d \log(tKM)),$$

$$\zeta_t(\delta) = \gamma_t(\delta) + \frac{\gamma_t^2(\delta)}{\sqrt{\lambda_t}} = \mathcal{O}(d \log(tKM)).$$

This implies that the following holds with probability at least $1 - \delta$:

$$||\theta - \theta^*||_{H_t} = \mathcal{O}(d \log(tKM))$$

for any $\theta \in E_t(\delta)$.  $\square$

# E  0.5 Approximation for Cascading Assortment Optimization

## E.1  Proof of 0.5 Approximation Ratio

Let $\phi(i, A) = \frac{w_i}{1+w(A)}$, where $w(A) = \sum_{i \in A} w_i$, which we use throughout this section. Recall that $N$ is the number of items in the ground set, and $K$ and $M$ denote the length of a cascade and assortment, respectively. Without loss of generality, $KM \leq N$ (if not, add dummy items with MNL weight 0).

For $M = 1$, we show that the problem is easy to solve optimally – simply pick the $K$ highest probability items and show them in arbitrary order.

**Lemma E.1.** *For general $M$, the optimization problem is weakly NP-hard even for $K = 2$.*

*Proof.* We use the hardness of unconstrained cascade optimization shown in [19] (Theorem 1). Given an instance of the unconstrained problem with $K = 2$ and ground set $[N]$, consider an instance of our cardinality constrained problem with $M = N$ over expanded ground set $[N] \cup [N]_0$ where $[N]_0$ consists of $|N|$ dummy elements that each have MNL weight parameter 0.  $\square$

**Lemma E.2.** *For any $M$, given a collection of assortments $\{A_1, \cdots, A_K\}$ with success probabilities $\{p_1, \cdots, p_K\}$, their order of display does not matter. Further, for every permutation $\rho : [K] \to [K]$, we have,*

$$\sum_{k \in [K]} p_k \prod_{\dot{k} < k}(1 - p_{\dot{k}}) = 1 - \prod_{k \in [K]}(1 - p_k) = \sum_{k \in [K]} p_{\rho^{-1}(k)} \prod_{\dot{k} < k}(1 - p_{\rho^{-1}(\dot{k})}).$$

*Proof.* If the customer views an assortment $A_k$, a success occurs in this assortment independently with probability $p_k$. We can (independently) pre-sample these Bernoulli random variables for each assortment. Then, the sequence in which assortments are shown does not matter since each ordering leads to the same end result (success of failure) once the Bernoulli variables are fixed. Algebraically, the probability that at least one of these random variables succeeds is given by $1 - \prod_{k \in [K]}(1 - p_k)$. An alternative way to compute these probabilities is to examine the random variables one by one until a success is found. If we examine the random variables in the order given by $\rho$, we get an alternative expression for the probability that at least one of the random variables succeeds, given by, $\sum_{i \in [K]} p_{\rho^{-1}(k)} \prod_{\dot{k} < k}(1 - p_{\rho^{-1}(\dot{k})})$. This completes the proof.  $\square$

**Lemma E.3.** *Let $\{A_k^*\}_{k \in [K]}$ denote the optimal solution. Then, $\cup_{i \in [K]} A_k^*$ is the set of $KM$ items with highest value of MNL weights.*

*Proof.* Suppose not. Then, there is an item $i$ in some assortment $A_k^*$ and an item $j$ that is not in any assortment such that $w_j > w_i$. Consider the assortment $A_k' = A_k^* \cup \{j\} \setminus \{i\}$. The probability of click is strictly higher in assortment $A_k'$ than in assortment $A_k^*$. Therefore, is we keep all other assortment as is but replace assortment $A_k^*$ with $A_k'$, we have a strictly better solution, contradiction.  $\square$

For $M = 1$, combining Lemma E.2 with Lemma E.3 shows that showing the $M$ highest probability items is optimal.

Now, order the items in $[N]$ in decreasing order of MNL weights and consider the following assortment for general $M$.

$$D_1 = \{1, 2, \cdots, M\}, D_2 = \{M + 1, M + 2, \cdots, 2M\} \cdots, D_K = \{(K - 1)M + 1, \cdots, KM\}.$$

Let OPT denote the overall click probability in the optimal solution.

**Lemma E.4.** *When $w(D_1) < 1$, we have,*

$$\text{OPT} \leq w(D_1) + w(D_2)\left(1 - w(D_1)\right) + \cdots + w(D_K) \prod_{k \in [K-1]} (1 - w(D_k)).$$

*Proof.* Given $w_1 + \cdots + w_M < 1$, we have. $w(D_k) < 1 \ \forall k \in [K]$. In fact,

$$\phi(A) := \sum_{i \in A} \phi(i, A) = \sum_{i \in A} \frac{w_i}{1 + w(A)} \leq w(A) \qquad \forall A \subseteq [N], |A| \leq M.$$

Given optimal solution $\{A_k^*\}_{k \in [K]}$, consider a hypothetical solution where for every $k \in [K]$, given that assortment $k$ is shown, the (independent) probability of click in assortment $k$ is $h_k := w(A_k^*)$ $(\geq \phi(A_k^*))$. We claim that this hypothetical solution, say $H$, has expected click probability at least as much as OPT. To see this, we couple the Bernoulli random variable for each assortment in $H$ with the corresponding assortment in OPT so that whenever there is a success in assortment $k$ in OPT, there is also a success in slab $k$ in $H$ (this is possible since $h_k \geq \phi(A_k^*)$). Thus,

$$H \geq \text{OPT}.$$

Now, it suffices to show that,

$$H \leq w(D_1) + w(D_2)(1 - w(D_1)) + \cdots + w(D_K) \prod_{k \in [K-1]} (1 - w(D_k)).$$

Observe that the RHS corresponds to a hypothetical solution with assortments $\{D_k\}_{k \in [K]}$ and success probabilities $\{w(D_k)\}_{k \in [K]}$ (instead of $\{\phi(D_k)\}_{k \in [K]}$). We now focus on the hypothetical scenario where the success probability of an assortment $A$ equals $w(A)$ (instead of $\phi(A)$). We show that in this scenario, the assortments $\{D_k\}_{k \in [K]}$ are optimal and this proves the main claim. We proceed by setting up a contradiction. Suppose that assortments $\{D_k\}_{k \in [K]}$ are sub-optimal and consider the optimal partition of $\cup_{k \in [K]} D_k$ into assortments $\{E_k\}_{k \in [K]}$. We have,

$$1 - \prod_{k \in [K]} (1 - w(E_k)) > 1 - \prod_{k \in [K]} (1 - w(D_k)).$$

Since assortments $\{E_k\}_{k \in [K]}$ are distinct from $\{D_k\}_{k \in [K]}$, there exists assortments $E_l$ and $E_n$ such that $w(E_l) > w(E_n)$ but $w_i < w_j$ for items $i \in E_l, j \in E_n$. Consider a new set of assortments $\{F_k\}_{k \in [K]}$ defined as follows,

$$F_k = \begin{cases} E_k & \forall k \in [K] \backslash \{l, n\}, \\ E_l \cup \{j\} \backslash \{i\} & k = l, \\ E_n \cup \{i\} \backslash \{j\} & k = n. \end{cases}$$

Observe that assortment $F_l$ has success probability $w(F_l) = w(E_l) + w_j - w_i$. Similarly, assortment $F_n$ has success probability $w(F_n) = w(E_n) - w_j + w_i$. Using Lemma E.2, let assortments $E_l, E_n$ and assortments $F_l, F_n$ be the last two assortments shown in their respective sequences. Then, the following inequalities show that the overall probability of success in $\{F_k\}_{k \in [K]}$ is strictly higher than the optimal partition $\{E_k\}_{k \in [K]}$, contradicting the optimality of $\{E_k\}_{k \in [K]}$.

$$\begin{aligned} 1 - (1 - w(F_l))(1 - w(F_n)) &= w(F_l) + w(F_n) - w(F_l)w(F_n), \\ &= w(E_l) + w(E_n) - (w(E_l) + w_j - w_i)(w(E_n) - w_j + w_i), \\ &> 1 - (1 - w(E_l))(1 - w(E_n)), \end{aligned}$$

here the last inequality follows from the fact that $w_j > w_i$ and $w(E_l) > w(E_n)$. $\square$

**Lemma E.5.** *The assortments $D_1, \cdots, D_K$, shown in any order, have overall click probability at least* $0.5 \, \text{OPT}$.

*Proof.* **Case 1:** Let $w_1 + w_2 + \cdots + w_M \geq 1$. Using Lemma E.2, we can show assortment $D_1$ first without loss of generality. Then, the probability of click in assortment $D_1$ is at least 0.5. Since probability of click overall is at least as much as the probability of click in $D_1$, we are done.

**Case 2:** Let $w_1 + \cdots + w_M < 1$. In this case, $1 - \sum_{i \in D_k} \phi(i, D_k) \leq 1 - 0.5 w(D_k)$ for all $k \in [K]$, and the overall click probability is at least,

$$1 - \prod_{k \in [K]} (1 - 0.5 \, w(D_k)).$$

Now, observe that,

$$1 - \prod_{k \in [K]} (1 - 0.5 \, w(D_k)) = 0.5 \, w(D_1) + \cdots + 0.5 \, w(D_K) \prod_{k \in [K-1]} (1 - 0.5 \, w(D_k)).$$

Comparing this term by term with the upper bound on OPT in Lemma E.4 completes the proof. $\square$

**Remark:** While the order of assortments does not matter here, if the customer was impatient and left early with some probability then the order would matter. In fact, in that setting it can be shown that displaying the slabs in the natural order $A_1, A_2, \cdots$ is 0.5 approximate.

# F  Limitations

While we study a more general version of combinatorial bandits, for the choice model, we adapt the MNL model. The MNL model is certainly one of the most popular options for modeling the outcomes of multi-class classification problems and certainly a practical and suitable extension of a simple linear model. However, it does have some drawbacks. For example, the MNL model relies on the Independence of Irrelevant Alternatives (IIA) assumption, and the utility functions in an MNL model are linear in parameters. In future work, we plan to address these challenges and extend to a more flexible choice model. However, for this work, we strongly believe that the current new model, proposed algorithms, and the regret analysis based on this newly proposed model provide more than sufficient contributions.

Consider MNL assortment bandits. Suppose a weight $w = w_1, \ldots, w_K \in \mathbb{R}^d$. For any given weight w, the expected reward function is as follows:

$$R(w) = \frac{\sum_{k \in [K]} \exp(w_k)}{1 + \sum_{k' \in [K]} \exp(w'_k)}.$$

At round $t$, the agent chooses the assortment $A_t = i_1, \ldots, i_K$. Let $w_t = x_{ti_1}^\top \theta_t, \ldots, x_{ti_K}^\top \theta_t$ and the optimal weight at round $w_t^* = x_{ti_1}^\top \theta^*, \ldots, x_{ti_K}^\top \theta^*$.

A second-order Taylor expansion gives that:

$$R(w_t^*) = R(w_t) + \nabla R(w_t)^\top (w_t^* - w_t) + \frac{1}{2}(w_t^* - w_t)^\top \nabla^2 R(\bar{w}_t)(w_t^* - w_t)$$

$$R(w_t^*) - R(w_t) = +\nabla R(w_t)^\top (w_t^* - w_t) + \frac{1}{2}(w_t^* - w_t)^\top \nabla^2 R(\bar{w}_t)(w_t^* - w_t)$$

$$R(w_t) - R(w_t^*) = -\nabla R(w_t)^\top (w_t^* - w_t) - \frac{1}{2}(w_t^* - w_t)^\top \nabla^2 R(\bar{w}_t)(w_t^* - w_t)$$

$$R(w_t) - R(w_t^*) = \nabla R(w_t)^\top (w_t - w_t^*) - \frac{1}{2}(w_t^* - w_t)^\top \nabla^2 R(\bar{w}_t)(w_t^* - w_t) \qquad (48)$$

Thus, if we apply Equation (28), the cumulative regret can be represented as follows:

$$\sum_{t=1}^{T} \{R(w_t) - R(w_t^*)\} = \sum_{t=1}^{T} \nabla R(w_t)^\top (w_t - w_t^*) - \frac{1}{2} \sum_{t=1}^{T} (w_t^* - w_t)^\top \nabla^2 R(\bar{w}_t)(w_t^* - w_t)$$

We first consider the upper bound of $\sum_{t=1}^{T} \nabla R(w_t)^\top (w_t - w_t^*)$ in the right side.

$$\nabla R(w) = \begin{bmatrix} R_1(w)R_0(w) \\ \vdots \\ R_K(w)R_0(w) \end{bmatrix}$$

where $R_k(w) = \frac{\exp(w_k)}{1 + \sum_{k' \in [K]} \exp(w'_k)}$ and $R_0(w) = \frac{1}{1 + \sum_{k' \in [K]} \exp(w'_k)}$