# OpenReview forum: "Cascading Contextual Assortment Bandits"
_NeurIPS.cc/2023/Conference — NeurIPS 2023 poster_

### Official Review · Reviewer_jQVq · 2023-07-01

**Soundness:** 2 fair
**Presentation:** 2 fair
**Contribution:** 3 good
**Rating:** 5
**Confidence:** 3

**Summary:**

This paper studies the contextual cascading assortment bandit problem, and proposes low regret algorithms for this problem.

**Strengths:**

1. This paper studies a novel problem by combining ideas from assortment bandits and cascading bandits.
	2. The paper introduces some new algorithmic ideas.

**Weaknesses:**

It seems to me the optimal cascading assortment should obey some structural properties, i.e., should similar items appear in the same assortment? Should different items appear in the same assortment? Should we pair desirable items together? Or should we pair high-weight and low-weight items together for an ‘anchoring effect’? It will be interesting to hear insights on this.

The writing contains noticeable grammatical errors and typos

The reward function in this work seems to greatly simplify the problem and removes the difficulty in prior work. See questions section below.

---------------
Update. Some issues in writing include:
1. several sentences in paragraph from line 215-227
2. line 272, the $Ht^{-1}$ I assume should be subscript.
3. Not writing out the constant in line 268 and 278 is strange.

**Questions:**

1. The first claim the paper makes is that they are able to remove the dependence on K in the cascading bandit problem by using a swapping technique. However, I believe this depends on the specific loss function used, therefore this claim is not entirely accurate. Specifically, the swapping technique only holds when the order does not matter, but this does not generally hold for cascading bandits.
2. The second claim is the regret removes $\kappa$ term. However, the regret definition is different from assortment bandits, therefore is this claim really accurate?
3. In section 2.4, is the description talking about the cascading bandit problem or cascading assortment bandit problem?

**Limitations:**

The author did not address limitations.

---

> ### Author Rebuttal · Authors · 2023-08-10
>
> Thank you very much for your time to review our paper and for your valuable feedback. We truly hope that we can resolve any doubts and misunderstandings of our result if there are any. Here are our responses to each comment and question.
>
> **Structural Properties of Optimal Cascading Assortment?**
>
> - The model is clearly defined in Section 2.2. Other than the MNL choice model structure stated in lines 127-128 and cascading interaction model, we do not impose any other structure on optimal assortment. With all due respect, we do not see why this has to be considered as a weakness.
>
> **"The reward function in this work seems to greatly simplify the problem and removes the difficulty in prior work"**
>
> - No, it does not. We strongly disagree with the comment. As we show in our paper, our proposed model encompasses two of the prominent existing combinatorial bandit instances, cascading bandits (K > 1, M=1) and assortment bandits (K=1, M>1), as well as single-action selection bandits (K=1, M=1), such as logistic bandits and multi-armed bandits with binary feedback. We propose a more general and more complicated model that is more challenging for regret analysis than the existing combinatorial parametric bandits. On top of that, we show even tighter and stronger regret bounds, overcoming the longstanding sub-optimal dependence!
>
>
> **Questions**
>
> **Q1: Swapping Technique and Dependence on $K$?**
>
> - You have stated that the swapping technique we propose has limitations in eliminating the dependency on $K$. Unfortunately, that is not correct. The swapping technique is intended to remove the worst-case scanning probability, denoted as $p^*$. The swapping technique has no relations to removing $K$ at all.
> - Now, we will show why dependence on $p^*$ appears and how we can mitigate this. See Section 4.2.1 in [16]. For contextual cascading bandits (with no assortments), the authors in [16] show that the regret at round $t$ is upper bounded as follows:
>     \begin{equation} \mathcal{R}^\alpha (t, S_t ) \le 2B \sum_{i \in S_ t } \beta_t (\delta) \|x_{ti}\|_{V_t^{-1}}. \end{equation}
> - Since $V_t$ contains only information of observed base arms, $i \in \lbrack O_\tau \rbrack: \forall \tau\in \lbrack t\rbrack$, there is out of control issue in summation. [16] copes with this issue by a resorting to the case that $O_{t} = |A_{tk}|$ using the worst-case scanning probability $p^*$ as below, which then results in $\frac{1}{p^*}$ dependence in the final regret bound in [16].
>     \begin{align}
>         \mathbb{E} \lbrack \mathcal{R}^\alpha (t, S_t ) \rbrack
>         \le \frac{1}{p^\star} \mathbb{E} \lbrack \mathcal{R}^\alpha (t, S_t ) \mathbb{1} \lbrace O_t =|S_t | \rbrace \rbrack
>         \le \frac{2B}{p^\star} \mathbb{E} \lbrack \sum_{i \in \lbrack O_t \rbrack } \beta_t (\delta) \|x_{ti}\|_{V_t^{-1}} \rbrack.
> 	\end{align}
> - However, in our case, the instantaneous regret at round $t$ is upper bounded as follows:
> \begin{equation*}
> \mathcal{R}^\alpha (t, S_t ) \le f(S_t , u_t ) - f(S_t , w_t^\star) \le 2\Big( \frac{K}{K+1}\Big)^{K+1} \max_{i \in S_t } \beta_t (\delta) ||x_{ti}||_{V_t^{-1}}$.
> \end{equation*}
> - As we mentioned in section 4.1 (line 242-244), the assortment with the largest item with respect to $\|x_{ti}\|_{V_t^{-1}}$ is always examined since it is in the first position of $S_t$ by our proposed swapping technique.
> - Due to the max operator and swapping technique, we can obtain the below inequality:
> \begin{equation*}
> \mathcal{R}^\alpha (t, S_t ) \le 2\Big( \frac{K}{K+1}\Big)^{K+1} \max_{i \in \lbrack O_t \rbrack} \beta_t (\delta)\|x_{ti}\|_{V_t^{-1}}.
> \end{equation*} Thus, $p^\star$ does not appear in our regret analysis.
>
> * The swapping technique is applied utilizing the structure of cacading model, where change in the order of assortments within the cascade does not affect the expected reward. Such a invariance is one of the key characteristics of the cascade model widely studied in the many existing previous literature [11, 12, 15, 21, 24].
>
> **Q2: "the regret definition is different from assortment bandits, therefore is this claim really accurate?""**
>
> - First, we would like to clarify the premise of your question that "the regret definition is different from assortment bandits." If you are referring to the use of  $\alpha$-approximate regret in our work (and in most cascading bandit literature) and the regret using exact optimization in assortment bandits, then we can set $\alpha = 1$, which is the regret using an exact optimization without approximation. Hence, regrets in our setting and assortment bandits (without cascade) are comparable when $K = 1$ which is a special case of our problem setting. Therefore, the claim is accurate.
> - Note that due to the complexity of computing the exact optimal solution in combinatorial optimization in general, it is very standard to use $\alpha$-approximate regret, denoted as $\mathcal{R}^{\alpha}$, in the wide range of combinatorial bandit literatures [7, 16, 22, 25, A, B]. To this end, in Appendix D, we even prove that an approximation solution using greedy algorithm for the cascading assortment optimization problem gives a 0.5 approximation of the optimal solution, which we believe is an independent contribution.
>
> **Q3: Is Section 2.4 about Cascading bandit problem or Cascading assortment bandit?**
>
> * The notions of $\alpha$-approximation oracle and $\alpha$-regret apply to our problem setting, cascading assortment bandit problem, which also includes both cascading bandits and assortment bandits. This is a very common notion in cascading and combiantorial bandit literature as seen in [7, 16, 22, 25, A, B] and many more.
>
> ---
>
> **References**
>
> [A] Wei Chen, Yajun Wang, and Yang Yuan. Combinatorial multi-armed bandit: General framework and applications. ICML, pp.151–159. PMLR, 2013.
>
> [B] Andi Nika, Sepehr Elahi, and Cem Tekin. Contextual combinatorial volatile multi-armed bandit with adaptive discretization. AISTATS, pp.1486–1496. PMLR, 2020.

---

> > ### Comment · Reviewer_jQVq · 2023-08-19
> >
> > Thank you for the response. However, I don't feel my questions were addressed. I will rephrase my questions.
> >
> > Q1. The reward function you are using is specified in line 134. The related work [12; 16] and [11] were cited. [11;12] considers the non-contextual setting, and weights can only be 0/1. The reward is conjunctive / disjunctive; in either case, the order in the cascade does not matter. [16] considers the contextual setting with a more general reward function. And as [16] observed, using their more general reward function, the order does matter in the cascade, affecting both the feedback and the reward.
> >
> > If I understand correctly, the reward function you are using follows more closely with [11;12], with the binary weight being replaced by the probability of being clicked, and reward is 1 if at least 1 item is clicked. Hence in your setting the order does not matter, and it is natural to move more uncertain items to the front as it aids parameter estimation (similar phenomenon has been observed in experiments in [11], where low preference items come early, as it helps learning).
> >
> > Now, it is claimed previous bounds depend on $K$ and this paper has improved this. I don't think this is entirely accurate for the following two reasons:
> > 1. In the non-contextual version, the complexity will depend on the number of arms $K$, this is unavoidable. In the contextual version, the parameter governing the complexity is $d$, the dimension, rather than the number of arms $K$. ( As an analogy, $K$ appears in multi-arm bandit regret bounds and $d$ appears in linear bandit regret bounds. )
> > 2. The paper is using a simpler reward function compared with [16].
> >
> > Q2. I understand the paper does not impose structural constraints on the cascading assortment. My question was, given the weights (assume latent $\theta$ is known), is there anything interesting we can say about the optimal solution (would similar weights be grouped together into an assortment, or different weights be grouped together)? Hence, my original question was more of a characterization question, not a learning question.
> >
> > To be clear, I do think the paper makes interesting contributions, I am just worried that the claims of improvement over prior work are not entirely accurate.

---

> > > ### Author Response · Authors · 2023-08-20
> > >
> > > Dear Reviewer jQVq,
> > >
> > > There seem to be fundamental misunderstandings in your comments. We genuinely hope that these can be addressed and resolved, and we approach this situation with an open attitude.
> > >
> > > To be candid, the communication with you has been particularly frustrating for us as authors. Apart from the fact that these comments have arisen at such a late stage, the fact that your reply came without an acknowledgment of any errors on your initial assessment, and with clear indications of misunderstanding about the basics of our problem setting, is disheartening. For instance, interpreting \(K\) as the total number of items (arms) rather than as the cascade length casts doubts on how effectively our work can be evaluated. We sincerely hope that these discrepancies are the result of unintentional mistakes.
> > >
> > > Despite the limited time remaining, our aspiration is to establish a foundation of shared understanding.
> > > Given the time constraints imposed by the impending deadline for the discussion period, our priority is to ensure that our dialogue rests on a common grasp of the basics. With this objective in mind, **we would greatly appreciate your input on the following straightforward *yes/no* questions** so that we both know we share some common grounds. This approach should facilitate a more productive discourse, given the brevity of time:
> > >
> > > 1. Do you acknowledge that **your initial assertion concerning the relationship between the swapping technique and the cascade length \(K\) is incorrect**? (Please note that the purpose of the swapping technique is to eliminate dependence on $p^*$. Please refer to our prior response for clarification.)
> > >
> > > 2. Is it clear to you that **\(K\) represents the cascade length in our work, and is not indicative of the total number of items**? (There seems to be confusion where \(K\) has been mistaken for the total number of items (arms), as indicated in your comment: “In the non-contextual version, the complexity will depend on the number of arms, this is unavoidable”)
> > >
> > > 3. Do you comprehend that **Li et al. 2016 [16] assume that the learning agent (algorithm) possesses precise knowledge of the position effect for each position**?
> > >
> > > Upon a positive response to all of the above questions, we can proceed with our subsequent discussions.
> > >
> > > ---
> > >
> > > **We respectfully disagree with your assertion that "the paper is using a simpler reward function compared with [16].**
> > >
> > > We kindly ask you to consider the substantial body of literature in the cascade bandits field that does not hinge on the assumption of **known** position effects, as employed in [16]. To illustrate this, we can readily cite numerous recent works [21, 24, C, D, E], among many others, even including in state-of-the-art results [Vial et al., 21]. It's pivotal to note that assuming *known position effects* is not universally recognized as standard practice, nor does it inherently denote technical advancement. Our approach adheres to the most prevalent form of cascading feedback that does not rely on the assumption of *known position effects*. Just because we do not use the assumption of "known position effects", should our work be considered simpler? We strongly dispute your claim.
> > >
> > > Whether one considers assumption of *known position effects* as [16] or not, our model considers *cascades of assortments* (subsets of multiple items per cascade), distinct from [16]'s focus on *cascades of single items*. (See Figure 1. [16] is the second figure "Cascading Bandit", and the our setting is the fourth one.) Furthermore, in [16], the click probability for each cascade (single item) is governed by a simple linear model, independently for each item. In contrast, our work employs the more intricate MNL choice model to compute the click probability for each assortment-based cascade which can accommodate substitution effect and correlation in click probabilities among items. As evident from the technical results and proofs presented in the supplementary material (which can be cross-referenced with the analysis in [16]), the regret analysis in our study is significantly more intricate. Notably, our work is the pioneering endeavor to explore cascades of MNL models for the first time.
> > >
> > > ---
> > >
> > > [21] Vial, Sanghavi, Shakkottai, and Srikant. "Minimax regret for cascading bandits." Advances in Neural Information Processing Systems 35, 2022.
> > >
> > > [24] Zhong, Chueng, and Tan. “Thompson sampling algorithms for cascading bandits.” The Journal of Machine Learning Research, 2021.
> > >
> > > [C] Kveton, et al. “On the value of prior in online learning to rank.” International Conference on Artificial Intelligence and Statistics. PMLR, 2022.
> > >
> > > [D] Zhong, Cheung, and Tan. “Best arm identification for cascading bandits in the fixed confidence setting.” International Conference on Machine Learning, 2020.
> > >
> > > [E] Wan, Ge, and Song. “Towards scalable and robust structured bandits: A meta-learning framework.” International Conference on Artificial Intelligence and Statistics, 2023.

---

> > > > ### Author Response · Authors · 2023-08-20
> > > >
> > > > ### Continuing from the previous comment
> > > >
> > > > **Dependence on \(K\) (cascade length) has recently been found to be avoidable in "non-contextual" cascading bandits [21]**. However, this issue has remained unresolved for contextual cascading bandits. A central pursuit in our paper is to address the fact that, while each reward is upper-bounded by a constant irrespective of cascade length, **previously established regret bounds have shown dependence on \(K\)** (even for the conventional cascade model without the assumption of *known position effects*).
> > > >
> > > > In essence, with or without *known position effects*, a suboptimal dependence on \(K\) persists, and eliminating such dependence has long been an open problem in the field of cascading bandits. Thus, the presence or absence of a position effect does not inherently contribute to solving the persistent issue of suboptimal \(K\) dependence. It's worth noting that our decision to forego the assumption of *known position effects* isn't driven by a desire to effortlessly eliminate \(K\) dependence, but rather because it is not conventional and might be practically far-fetched (e.g., how would one ascertain exact position effect values for each position beforehand in real-world scenarios?).
> > > >
> > > > [It's important to highlight that dependence on \(N\) (total number of items), not \(K\) (cascade length), can be circumvented through straightforward parametrization—whether employing a linear, logistic, or MNL model. Obviously, in non-contextual settings, \(N\) dependence is inevitable due to the absence of generalization. It's worth clarifying that our focus is not on eliminating \(N\) dependence, as this is automatically achieved with the parametric assumption, and we trust you recognize this distinction. With all due respect, we do not understand why you would even mention this \(N\) dependence, even if you are mistaken. If you mistook \(K\) for \(N\), how does that support your assessment of our contributions?]
> > > >
> > > > We formally establish the order invariance property of the conventional cascade model in Lemma 2.1 (this property applies not only to our cascade of assortments but also to all conventional cascade models without position effects). Please understand that this is not an attempt to simplify the model, as you appear to persistently argue. Rather, it's a rigorous demonstration of a fundamental characteristic of the conventional cascading model, which is widely used. Our intent is to potentially leverage this property in designing algorithms. **However, it's important to reiterate that the order invariance property has no effect on eliminating \(K\) dependence**. Even then, we are NOT assuming this property. We prove this result!
> > > >
> > > > ---
> > > >
> > > > **Regarding your query on optimal assortments**
> > > >
> > > > First, we do not understand your usage of "similar weights" and "different weights". The weights $w_{ti}$—whether referring to true or estimated—are 1-dimensional values in $\mathbb{R}$. How do you define similarity?
> > > >
> > > > Let us elaborate more on the optimal assortment. Please note that we investigate the assortment optimization problem in Appendix D. We show that if the MNL weights are known the problem is weakly NP-hard to solve even with K=2. The reduction uses the hardness of partition-type problems which indicates that the optimal cascade may not be as simple as similar or dissimilar items together but may involve a careful mixture of some higher-weight items in assortments of the cascade. Note that we give a fast and simple 0.5 approximation to the problem.  The algorithm selects higher-weight items and groups them into different assortments, not necessarily in a top-bottom order. In the absence of further optimization assumptions (such as optimization constrained to exhibit diversity, then one first needs to define what diversity means, etc.), our solution would be generic.

---

> > > > ### Comment · Reviewer_jQVq · 2023-08-20
> > > >
> > > > Ok, my main concern was the use of the rather restrictive disjunctive objective, but as the authors pointed out, this still remains the popular choice among state-of-the-art works, I will adjust my score.

---

> > > > > ### Author Response · Authors · 2023-08-21
> > > > >
> > > > > Thank you for your response and re-evaluation of our work. We have also noticed the additional comments you've provided recently in the original review regarding writing. We would like to direct your attention to the fact that the supplementary version includes an improved main text. In this revised version, several of the aspects you've mentioned have been addressed and improved upon. Your consideration of improved writing would be greatly appreciated. Also, note that we have included the limitation section in Appendix E. Thanks.

---

### Official Review · Reviewer_Q2NZ · 2023-07-05

**Soundness:** 3 good
**Presentation:** 4 excellent
**Contribution:** 3 good
**Rating:** 7
**Confidence:** 3

**Summary:**

This paper studies a new contextual combinatorial multi-armed bandit model, which generalizes the contextual cascading bandits and assortment bandits.  For the offline problem when item parameters are known, the authors propose a 0.5-approximate solution. For the online problem where parameters are not known a priori, the authors first propose a UCB algorithm, called UCB-CCA, which yields a regret bound of $\tilde{O}(\kappa^{-1}d\sqrt{T})$ regret bound. To remove the unsatisfying $\kappa$ which may be relevant to the cascade length $K$, the authors further leverage Bernstein-type concentration and propose a new algorithm UCB-CCA+, which removes the $\kappa^{-1}$ dependence and achieves regret bounds that are independent of $K$. Finally, the authors conduct experiments to show the practical efficacy of the proposed methods.

**Strengths:**

Overall, I feel this is a decent work that is suitable to put in the combinatorial MAB literature.

1. From the model perspective, the model is new and general, which covers contextual cascading bandits and MNL bandits as degenerate cases.
2. For the results, Table 1 gives a clear comparison with existing works and this paper gives the first regret bound for this new model. Interestingly, when the
3. For the analysis, this paper not only is grounded by existing works from MNL bandit, but also gives some new techniques
4. For the writing, it is clear and intuitive supported by intuitive figures and tables.

**Weaknesses:**

Overall, I do not have major concerns, yet I have some minor comments, which I hope to get some clarification to validate my understanding.

1. In line 8 of Algorithm UCB-CCA+, the algorithm uses the true parameter $w_t^*$ which is unknown, is it a typo?
2. In line 4 of Algorithm UCB-CCA+, it is a combinatorial optimization problem over a confidence radius $B_t(\delta)$, it is NP-hard in general? Is any computational-efficient method for this?
3. I cannot find the lower bound result for the current problem. Is the current result matches the lower bound?

**Questions:**

Please comment point 1,2,3 in the above weakness part.

**Limitations:**

The authors adequately addressed the limitations and there are no potential negative societal impact of their work.

---

> ### Author Rebuttal · Authors · 2023-08-09
>
> Thank you very much for your time to review our paper and for your valuable feedback. Here are our responses to each comment and question:
>
>
> **Typo in Algorithm UCB-CCA+**
>
> * Yes, it is a typo. Thank you very much for catching it. It should be corrected to $\theta_t$.
>
> **Combinatorial Optimization in UCB-CCA+**
>
> - Thank you for your question. Yes, finding the optiaml cascade is weakly-NP hard as we show in Lemma D.1 in the appendix. To this end, in Appendix D, we prove that a greedy selection for the cascading assortment optimization problem gives a 0.5 approximation of the optimal solution. We believe such approximation optimization guarantee is very rarely shown in combinatorial bandit literature. We believe that this result serves an independent contribution.
>
> **On Lower Bounds**
> - Thank you for your questions on possible lower bounds. For logistic bandits ($K=1, M=1)$ which is a special case of our problem setting, [A] established a regret bound as $\Omega(d\sqrt{T})$. Also, [5] proved that a regret lower bound for assortment MNL bandits ($K=1, M\geq1$) is $\Omega(d\sqrt{T})$. Thus, our regret upper bound matches with these lower bounds in terms of time horizon $T$ and dimensionality $d$ in these special cases.
> - Lastly, for non-contextual cascade bandits, [21] derived a regret lower bound of $\Omega(\sqrt{LT})$ where $L$ is the total number of items, which does not depend on the cascade length $K$. Hence, the $K$-indepedence in the regret upper bound in our result appears to be sound and tight in terms of $K$.
> - For general contextual cascading assortment bandits ($K>1, M>1), to our knowledge, proving a regret lower bound remains an open problem. We will include these discussions on lower bounds in a revised version of our paper.
>
>
> ---
>
> **References**
>
> [A] Marc Abeille, Louis Faury, and Clément Calauzènes. "Instance-wise minimax-optimal algorithms for logistic bandits." International Conference on Artificial Intelligence and Statistics. PMLR, 2021.

---

> > ### Comment · Reviewer_Q2NZ · 2023-08-18
> >
> > Thanks for the clarification from the authors. I am still a little confused about the computation problem of UCB-CCA+. I think your 0.5-approximate solution can only apply when you input the (optimistic) weights, e.g., line 3 of UCB-CCA has the $u_{t,i}$. However, for UCB-CCA+, the confidence radius is over $g_t(\theta)$, where you cannot have an explicit form of $u_{t,i}$ like line 3 of UCB-CCA. Therefore, in line 4 of UCB-CCA+, you need to have a double-oracle that optimizes over $\theta$ and $S$, which should be NP. Please correct me if I am wrong, e.g., you are actually using Eq. (7) to compute $u_{t,i}$ for UCB-CCA+.

---

> > > ### Author Response · Authors · 2023-08-19
> > >
> > > Thank you for your question, and we are more than happy to elaborate further on the optimization in MNL-CCA+. As you noted, one would have to resort to a joint optimization oracle in general. But, there are also ways to utilize the approximate optimization result. For any given parameter $\theta$, we can compute a 0.5 approximate cascade optimal with respect to the given parameter by using the greedy algorithm. The analysis of the greedy algorithm (Lemma D6 in particular) also indicates that we can replace the objective $f(S,w_t)$ with a simpler proxy function. Also, the approximate guarantees can be applied not only to UCB weights $u_t$ but also to any given weights $w_t$ -- the solution would be approximately optimal with respect to those particular weights being used. However, searching for the optimal $\theta$ is hard since the set $B_t(\delta)$ may be non-convex -- this is also evident in the previous literature in assortment bandits (e.g., Agrawal et al. 2023 [2]). Based on this, a possible way to compute an approximate assortment is as follows. We could use a grid search heuristic that searches over a grid of points in the set $B_t(\delta)$. for each point $\theta$ in the grid, we use the greedy algorithm to obtain a 0.5 approximate cascade and finally, compare all the candidate cascades and choose the best one among them. Again, we appreciate your question and constructive feedback. If you have any further questions, please feel free to let us know.

---

> > > > ### Comment · Reviewer_Q2NZ · 2023-08-20
> > > >
> > > > Thanks for the reply from the authors. I encourage the authors to include this discussion of this joint optimization as a remark in the main paper. Also as noted by the reviewer anEU, you can add the discussion with [A] regarding contextual cascading bandits to increase the timeliness of the current work.
> > > >
> > > > Besides the above two suggestions, I do not have further concerns and would like to keep my score unchanged.
> > > >
> > > > [A] Xutong Liu, Jinhang Zuo, Siwei Wang, John CS Lui, Mohammad Hajiesmaili, Adam Wierman, and Wei Chen. "Contextual Combinatorial Bandits with Probabilistically Triggered Arms." In International Conference on Machine Learning, 2023.

---

> > > > > ### Author Response · Authors · 2023-08-20
> > > > >
> > > > > Thank you for your constructive feedback. Yes, we will incorporate your suggestions in our revision. Thank you very much for recognizing the value of our work and your support!

---

### Official Review · Reviewer_tk8N · 2023-07-09

**Soundness:** 3 good
**Presentation:** 3 good
**Contribution:** 3 good
**Rating:** 5
**Confidence:** 3

**Summary:**

This paper studies the Cascading Contextual MNL bandits problem. Two effective algorithms UCB-CCA and UCB-CCA+ are proposed. Compared to existing cascading bandits and MNL bandits, the regrets of the two algorithms have some better dependence on the length of cascades and \kappa. Numerical simulations demonstrate the effectiveness of the proposed algorithms.

**Strengths:**

S1. Combining cascading bandits and MNL bandits is interesting and can find real applications.

S2. The proposed UCB-CCA algorithm has a regret independent of K, the length of cascades.

S3. The results of numerical simulations are good.

**Weaknesses:**

W1. This paper focuses on removing the dependence on K and \kappa. It seems to me that improving the dependence on d may be more helpful than removing the dependence on K, since in practice K could be a small constant and the dimension of contextual vectors could be large. Note that [18] has an algorithm for MNL contextual bandits that has a \sqrt{dT} regret.

W2. In theorem 5.2, the dependence on \kappa is removed by increasing T. Such a treatment explicitly assumes that T is much larger than 1/\kappa, which makes some sense in practice. However, I am not sure if this practical assumption is appropriate in the theoretical analysis of regret.

W3. In the numerical simulations, the authors report the cumulative regrets of algorithms. What about the curves of revenues of algorithms? Since all algorithms adopted in experiments have sublinear regrets, the cumulative regret is actually an insignificant term compared to the revenue. I wonder if the difference between the proposed algorithms and the baseline is still as large as reported in Fig 2 when reporting the curves of cumulative revenues.

W4. The submission file and the full version (Supplementary Material) are inconsistent with each other in some parts.

**Questions:**

Q1. For UCB-CCA+, as \kappa is removed in the regret, does it mean that we only need the assumption that the Fisher information matrix is invertible?

**Limitations:**

The authors discuss the limitations of their work in Appendix.

---

> ### Author Rebuttal · Authors · 2023-08-09
>
> Thank you very much for your time to review our paper and for your valuable feedback. Here are our responses to each comment and question:
>
>
> **improving dependence on $d$?**
>
> - We believe that you are referring to $\tilde{\mathcal{O}}(\sqrt{dT})$ regret in Theorem 4 of [19]. Then, please note that the regret bound in Theorem 4 of [19] contains $\log(TN)$ dependence. That is, if the total number of items is very large, such that $N>\exp(d)$, then the regret bound would eventually be even worse than $\tilde{\mathcal{O}}(d\sqrt{T})$.
> - Note that in both logistic [A] and MNL bandits [5] which are both special cases of the cascading contextual assortment bandits, the regret lower bound is of $\Omega(d\sqrt{T})$. Hence, this suggest that $\mathcal{O}(d)$ dependence cannot be improved for arm-independent bounds. Note that our regret upper bounds in Theorem 4.1 and Theorem 5.2 are in the regime of arm-independent bounds. Hence, we do not see why this has to be considered a weakness.
>
> **$T$ and $1/\kappa$**
>
> - First of all, note that Theorem 5.2 holds true for all values of $T$, regardless of whether $T$ is larger than $1/\kappa$ or not. Hence, there is **no necessity for the assumption of $T \gg 1/\kappa$ at all** in order for Theorem 5.2 to hold. It is just that depending on the relationship between $T$ and $1/\kappa$, the leading term may be different.
> - Now, suppose $T$ is small enough such that the second term in Theorem 5.2 becomes dominant as you stated. Then, the total regret would be $\mathcal{O}(\frac{1}{\kappa} d^2 \log T)$ which in this case has $\frac{1}{\kappa}$ dependence but  is logarithmic in $T$, and we have already assumed that $T$ is small if the second term were to be a learning term. So, for small $T$, $\log T$ becomes even smaller. Hence, such a case is not a concern, both theoretically and practically. Hence for sufficiently large $T$, the regret is $\mathcal{O}(d \sqrt{T})$ and small enough $T$, $\mathcal{O}(\frac{1}{\kappa} d^2 \log T)$.
>
>
> **Plotting curves of revenues?**
>
> - Considering that the definition of regret is the cumulative difference between the expected reward of the optimal action and the expected reward of the action chosen by the agent, conceptually and for numerical purposes, we do not see any difference between comparsion based on regret and comparsion based on revenue. Given that standard metric in the bandit literature by default is regret, we do not see why this has to be considered a weakness. We would be more than happy to include plots in terms of revenue if required.
>
>
> **Supplementary Material**
>
> - When the supplementary material was submitted, the entire manuscript was uploaded which included the main text for convenient reading that includes the proofs and hyperlinks within the document. The uploaded supplementary material includes minor revisions in the main text.
>
> #### Questions
>
> **Q1: Assumption on the Fisher information matrix only?**
>
> * We still need Assumptions 2.2 and 2.3. What we claim is that $\kappa$ in Assumption 2.3 no longer depends on the leading term of our regret upper bound.
>
>
> We trust that our responses have sufficiently addressed your questions and alleviated any concerns. Should you need any clarification, we are more than happy to address them during the discussion period.
>
> ---
>
> **References**
>
> [A] Marc Abeille, Louis Faury, and Clément Calauzènes. "Instance-wise minimax-optimal algorithms for logistic bandits." International Conference on Artificial Intelligence and Statistics. PMLR, 2021

---

> > ### Comment · Area_Chair_mUaa · 2023-08-21
> > **further comments to the authors' rebuttal?**
> >
> > Dear Reviewer tk8N,
> >
> > Do you have further comments on the authors rebuttal?
> >
> > Area Chair

---

### Official Review · Reviewer_anEU · 2023-07-13

**Soundness:** 2 fair
**Presentation:** 3 good
**Contribution:** 3 good
**Rating:** 6
**Confidence:** 4

**Summary:**

This paper studies a new combinatorial bandit problem that generalizes the existing cascading and assortment bandits. The authors first propose a UCB-based algorithm, UCB-CCA, that achieves a tighter regret bound than existing bounds for cascading contextual bandits by eliminating the dependence on cascade length $K$. They also introduce an improved algorithm, UCB-CCA+, and use a Bernstein-type concentration to prove a regret bound without $\kappa^{-1}$ dependence, where $\kappa$ is a problem-dependent constant in the regret bound of UCB-CCA. Numerical experiments validate the effectiveness of the proposed algorithms.

**Strengths:**

1) This paper is the first to study the combination of contextual cascading and assortment bandits. This new problem is well-motivated by real-world applications in recommender systems.
2) One of the main technical contributions is the new Lipschitz continuity of the expected reward function for contextual cascading assortment bandits in Lemma 4.2, which helps prove the regret bound of UCB-CCA is independent of $K$ and $M$. (However, I have a question about the proof of this Lipschitz continuity; see below.)
3) The proposed UCB-CCA+ algorithm achieves an improved regret bound than that of UCB-CCA, solving the two technical challenges (dependence on cascade length and $\kappa$) faced by contextual cascading and assortment bandits simultaneously.

**Weaknesses:**

1) For contextual combinatorial bandits, there is a recent result [A] that provides a regret bound independent of the cascade length $K$ using a variance adaptive algorithm. Moreover, its regret bound can get rid of $p^*$, which raises a concern that whether the optimistic exposure swapping in Section 3.3 is necessary or whether the $p^*$ issue can be resolved by a more involved analysis.
2) The Lipschitz continuity in Lemma 4.2 is a key component of the analysis. However, the proof in line 402-405 is unclear to me. I would appreciate it if the authors could add more details about the proof; a simple example of the non-contextual cascading bandit can also be helpful.
3) Although UCB-CCA+ achieves a good regret bound, there is no discussion on the lower bound of contextual assortment combinatorial bandits: would it be similar to that of the contextual combinatorial bandits or contextual assortment bandits?

[A] Xutong Liu, Jinhang Zuo, Siwei Wang, John CS Lui, Mohammad Hajiesmaili, Adam Wierman, and Wei Chen. Contextual Combinatorial Bandits with Probabilistically Triggered Arms. In International Conference on Machine Learning, 2023.

**Questions:**

Typo: line 207: In round (at every round) $t$

**Limitations:**

See Weaknesses.

---

> ### Author Rebuttal · Authors · 2023-08-09
>
> Thank you very much for your time to review our paper and for your valuable feedback. Here are our responses to each comment and question:
>
> **Comparison with [A]**
>
> *  Thank you for introducing [A]. We are more than happy to compare our work with [A].
> *  First of all, we would like to point out the NeurIPS policy on "recent work" which states, *"What is the policy on comparisons to recent work? Papers appearing less than two months before the submission deadline are generally considered concurrent to NeurIPS submissions. Authors are not expected to compare to work that appeared only a month or two before the deadline."*
> (https://neurips.cc/Conferences/2023/PaperInformation/NeurIPS-FAQ#:~:text=are%20generally%20considered%20concurrent%20to,or%20two%20before%20the%20deadline.)
> - [A] was published in ICML in July 2023, and its first arXiv version was posted on March 30th, 2023, which was about a month and a half before the NeurIPS submission deadline. Hence, we are not obliged to compare [A] with our work by the policy. Nevertheless, we are more than willing to offer a comparison.
>
> * Upon reviewing [A], we observed that there are significant technical differences in the analysis between our work and [A]. To elaborate the distinction, writing the expected reward function as $\sum_{i=1}^{K}p_{i}^{\mu, S}(\bar\mu_{i} - \mu_{i})$ in [A], where $p_{i}^{\mu, S} = \prod_{j=1}^{i-1}\mu_{j}$, allows them to remove $p^\star$. Then, the difference $\bar\mu_{i} - \mu_{i}$ is represented as a weighted norm of the feature vector, $||x_{i}||_{V_t^{-1}}$. To square a weighted norm of the feature vector, the Cauchy-Schwarz inequality can be applied and a dependency on $K$ arises. That is, the regret bound in Theorem 1 of [A] is independent of $p^\star$, but dependent on $K$. Improving upon the $K$-dependent regret bound, [A] shows that contextual cascading bandits satisfy triggering probability and variance modulated (TPVM) condition by Lemma 19 in [B] and eventually carves off the $K$ dependence.
> * On the other hand, in our work, to simultaneously eliminate dependencies on both $p^\star$ and $K$, we utilize the mean-value theorem combined with the swapping technique. We observe that the techinques used in both works are unique. Also, we would like to clearly highlight that our model is based on MNL choice model (and considers cascades of assortments) whereas [A] is based on much simpler linear click model.
>
> **Lipschitz continuity in Lemma 4.2**
>
> * By the mean value theorem, \begin{align*}
>         f(S_t , u_t ) - f(S_t , w_t^* ) = \nabla_\theta f(S_t ,\bar w)(\theta_t - \theta^* )
>         = \left\lbrace \prod_{A_{t \dot{k}}\in S_t } p_t (i_0 |A_{t \dot{k}}, \bar w) \right\rbrace \sum_{A_{tk}\in S_t } \sum_{i \in A_ {tk}} p_t (i|A_{tk}, \bar w) x_{ti}^\top (\theta_t - \theta^* )
>         \end{align*}
> * For a convenience, let $\sum_{i\in A_ {tk}}p_{t}(i|A_{tk}, \bar{w}) := P_{tk}$. Then, we can simplify $\lbrace \prod_{A_{t\dot{k}}\in S_{t}} p_{t}(i_{0}|A_{t\dot{k}}, \bar{w}) \rbrace \sum_{A_{tk}\in S_{t}} \sum_{i\in A_ {tk}}p_{t}(i|A_{tk}, \bar{w})$ as follows: $\prod_{\dot{k}\in[K]}(1 -P_{t\dot{k}})\sum_{k\in[K]}P_{tk}$.
> * We can see that this expression is maximized as $\left(\frac{K}{K+1}\right)^{K+1}$ when $P_{tk}=\frac{1}{K+1}$ for all $k\in[K]$, since $0 < P_{tk} < 1$.
>
> **On Lower Bounds**
> - Thank you for your questions on possible lower bounds. For logistic bandits ($K=1, M=1)$ which is a special case of our problem setting, [C] established a regret bound as $\Omega(d\sqrt{T})$. Also, [5] proved that a regret lower bound for assortment MNL bandits ($K=1, M\geq1$) is $\Omega(d\sqrt{T})$. Thus, our regret upper bound matches with these lower bounds in terms of time horizon $T$ and dimensionality $d$ in these special cases.
> - Lastly, for non-contextual cascade bandits, [21] derived a regret lower bound of $\Omega(\sqrt{LT})$ where $L$ is the total number of items, which does not depend on the cascade length $K$. Hence, the $K$-indepedence in the regret upper bound in our result appears to be sound and tight in terms of $K$.
> - For general contextual cascading assortment bandits ($K>1, M>1), to our knowledge, proving a regret lower bound remains an open problem. We will include these discussions on lower bounds in a revised version of our paper.
>
> ---
>
> **References**
>
> [A] Xutong Liu, Jinhang Zuo, Siwei Wang, John CS Lui, Mohammad Hajiesmaili, Adam Wierman, and Wei Chen. "Contextual Combinatorial Bandits with Probabilistically Triggered Arms."" In International Conference on Machine Learning, 2023.
>
> [B] Xutong Liu, Jinhang Zuo, Siwei Wang, Carlee Joe-Wong, John Lui, and Wei Chen. "Batch-size independent regret bounds for combinatorial semi-bandits with probabilistically triggered arms or independent arms."" Advances in Neural Information Processing Systems, 35:14904–14916, 2022.
>
> [C] Marc Abeille, Louis Faury, and Clément Calauzènes. "Instance-wise minimax-optimal algorithms for logistic bandits." International Conference on Artificial Intelligence and Statistics. PMLR, 2021.

---

> > ### Comment · Reviewer_anEU · 2023-08-18
> >
> > Thanks for the detailed response. It addresses most of my concerns. One more note on the comparison with [A]: from my understanding, the algorithm in [A] can get rid of $p^*$ and $K$ simultaneously according to their Table 2 (both Disjunctive and Conjunctive Combinatorial Cascading Bandits satisfy the TPVM condition). I would like to maintain my score.

---

> > > ### Author Response · Authors · 2023-08-18
> > >
> > > We are glad our responses have addressed your concerns. Thank you very much for your support and overall positive feedback.

---

### Official Review · Reviewer_nP45 · 2023-07-26

**Soundness:** 2 fair
**Presentation:** 3 good
**Contribution:** 3 good
**Rating:** 6
**Confidence:** 3

**Summary:**

This paper introduces the cascading contextual assortment bandit problem and provides a UCB type algorithm. This problem is motivated by online content recommendation systems. They develop a UCB algorithm that is applicable to this problem setting, and prove that their algorithm improves upon existing regret bound rates in the cascading and assortment bandit problems respectively.

**Strengths:**

- Based on the authors' discussion, the scaling rate of their regret bounds is both sharper and more interpretable/intuitive than that in existing literature. It seems their regret bound improves upon those both in the assortment and cascading bandit literatures respectively.
- The authors have a nice discussion about why we expect the regret to decrease with $K$ and then show how their result shows this type of dependence. Additionally, their result and discussion of how their regret bound scales with $\kappa$ in a way that is not worsening its dependency on $M$ seems nice.
- The overall writing and presentation in the paper pretty good. I think the table 1 is quite useful and Figure 1 was very helpful for understanding the problem.


**Weaknesses:**

- In the evaluation, it was not clear to me why you only compared to C^3-UCB and not the other methods listed in table 1. While the algorithm has these regret guarantees, its not entirely clear if the algorithm really performs well in practice from the simulations in the evaluation section.
- Only applicable to environments in which the probability that a user clicks is determined by a generalized linear model. Additionally, the feature vector $x_{ti}$ captures both contextual information on the user and the item. It's not clear how this kind of vector could be chosen in practice.
- There is not a real related works section in the main paper. I see there is one in the Appendix.

**Questions:**

- It is not clear to me that Assumption 2.3 as written will ever hold. Do you really need to take an infinum over all $\theta \in \mathbb{R}^d$? Based on my understanding, this means that if you take $\theta = \lambda \cdot x_i$ you can make $w_i = x_i^\top \theta$ arbitrarily small or large with the choice of $\lambda$. Then, based on the model from line 127, it seems you could make $p_t(i_m | A_{tk}, w)$ arbitrarily small. Can you explain there any reasonable settings where Assumption 2.3 will hold? If there are, please have a discussion of this and also more information about how to interpret Assumption 2.3.
- You say below assumption 2.2 that the regret bound is $c$ times larger if you allow the norm of $x_i$ and $\theta^\star$ respectively to be bounded below $c$. Is \emph{knowledge} of $c$ needed by the algorithm? In other words, does your algorithm currently implicitly take advantage of knowledge that you assume that $x_i$ and $\theta^\star$ are bounded by $1$? If so, this would severely limit the practical applicability of the approach. If this is the case, this limitation should be discussed. If it is not, it should also be mentioned when discussing scaling by $c$.
- Could you a sentence or two (or more) about how you suggest to choose ridge penalty $\lambda$?

**Limitations:**

- I would like the authors to address the questions / limitations I list in the weaknesses section.

---

> ### Author Rebuttal · Authors · 2023-08-09
>
> Thank you very much for your time to review our paper and for your valuable feedback. Here are our responses to each comment and question:
>
> **Numerical Evaluations**
>
> - First of all, since our problem setting and proposed model, the cascading assortment bandit, are novel, there are not existing methods proposed exactly under this new model. Hence, we can only report to comparing against special cases. While both assortment bandits and cascading bandits are special cases of our model, assortment bandits do not possess cascading effect for which removing the suboptimal dependence on the cascade length is one of the main objectives. Hence, we aim to compare with cascading bandit algorithms to see the effect. Among the contextual cascading bandit algorithms, C$^3$-UCB is one of very few algorithms whose implementation exist. In this rebuttal (see the pdf file attached to the global rebuttal), we have included a comparison with another related method, CombCascade in [12], and we plan to incorporate further comparisons with other methods in a revision. We would be happy to include more results in the list. We appreciate your feedback in making our paper more persuasive to readers through additional experimental performance comparisons. But also, as this is the first theoretical work proposing the new cascading assortment bandit, the new provably efficient algorithms, and their improved regret bounds removing suboptimal dependence that existed even in less general cases, we respectfully request that it be mainly assessed based on its theoretical merit.
>
>
> *"Only applicable to environments in which the probability..."*
>
> - All parametric models for clicks in the bandit framework (whether it is a linear, logistic, or MNL model, along with their combinatorial adaptation, such as cascading, assortment, semi-bandit, etc.) have their own modeling assumption on click probability. Regret bounds are mostly derived under the realizability of each modeling assumption. Hence, we do not necessarily agree with the comment that "applicability to environments" with modeling assumptions should be considered a weakness. Rather, as we show in our paper, our proposed model encompasses two of the prominent existing combinatorial bandit instances, cascading bandits (K > 1, M=1) and assortment bandits (K=1, M>1), as well as single-action selection bandits (K=1, M=1), such as logistic bandits and multi-armed bandits with binary feedback. Under this more general model, we show even tighter and stronger regret bounds!
>
>
> *"the feature vector $x_{ti}$ captures both contextual information on the user and the item..."*
>
> - Suppose the user at round $t$ is characterized by a feature vector $u_t$ and the item $i$ has a feature vector $v_{ti}$ (note that we can allow item feature to vary over time), then we can use context feature vector as $x_{ti} = \text{vec}(u_t v_{ti}^{\top})$, the vectorized outer-product of $u_t$ and $v_{ti}$, as the combined feature vector of item $i$ at round $t$. This is a common technique also used in [16, 18, 19]. If a user's information is not accessible (for example, due to privacy issues), then one can use item-dependent features only, say $x_{ti}=v_{ti}$.
>
>
>
> **Related Works Section**
>
> - Due to the limited space in the main text, we defer the "Related works" section to the appendix. We are more than happy to moving the "Related works" to the main text in the revisied version.
>
>
>
> **Questions**
>
> **Q1 on Assumption 2.3**: It is a very good question. First of all, Assumption 2.3 is the standard regularity assumption in the MNL contextual bandit literature [5, 6, 18, 19, 20, 23]. Since true $\theta^*$ is assumed to be $\|\theta^*\| \leq 1$ (in Assumption 2.2), we can only consider $\theta \in \mathbb{R}^d$ with $\|\theta\| \leq 1$ for Assumption 2.3 (hence modification in the subscript). Since $\|x_i\| \leq 1$ for all $i$, $x_i^\top \theta$ is bounded by $-1 \leq x_i^\top \theta \leq 1$ for all $i$. Hence, the probability $p_t(i | A_{tk}, w)$ cannot be arbitrarily small. Now, one practical implication of this assumption is that under any choice model we can possibly consider, we consider items that provide utilies to users (items that has aleast some probilities to be clicked).
>
> **Q2 on Assumption 2.2**: The boundness assumption is also a standard assumption in almost all parametric bandit literature [1,3,5,14,16,18,19,20,23] that includes linear, logistic, GLM, and MNL bandits. However, in practice and also in theory, you do not have to tune $c$ seperately. Since all the unknown hyperparameters can be combined $c' := c \cdot \frac{1}{\kappa} \cdot \sigma$ and tuned as a whole (where $\sigma$ is sub-gaussian parameter which is assumed in almost all parametric bandits; note that in MNL bandits, it is known that $\sigma = \frac{1}{2}$ but in general parametric bandits, $\sigma$ is not known). If this were to be considered hinderance, then the same argument should be made about linUCB and linTS as well as almost all existing parametric bandit algorithms.
>
> **Q3 on Choosing $\lambda$**: If $\lambda$ can be any value between 1 and $d$, then the regret bound would not change the leading factor. Hence, a common choice of $\lambda$ is $\lambda = 1$ or $\lambda = d$.

---

> > ### Comment · Reviewer_nP45 · 2023-08-10
> > **Response**
> >
> > Thank you for your response.
> >
> > Regarding Assumption 2.3, will you revise Assumption 2.3 to only consider $\theta \in \mathbb{R}^d$ such that $\\| \theta\ \| \leq 1$? Or are you saying that as stated in the paper, this is the assumption you need? My understanding is that currently the Assumption 2.3 as stated is too strong and not likely to hold, but you actually don't need it to be so strong for your proofs, as you just need an infinum over $\theta \in \mathbb{R}^d$ such that $\\| \theta \\| \leq 1$.
> >
> > Regarding Assumption 2.2, I think it is okay to have this limitation, but I think you should be up front about it. I suggest you add a statement about it and also mention that this limitation is also true of many other common algorithms in the literature. This will help the future readers of your paper, who may want to apply your approach understand the strengths and weaknesses of your method.

---

> > > ### Author Response · Authors · 2023-08-10
> > >
> > > Thank you. Yes, we will revise Assumption 2.3 to include $\lVert \theta \rVert \leq 1$. Of course, we are more than willing to be up front about all our assumptions as we already stated the scalability of the upper bound on norms. We can include more discussion on Assumption 2.2. Thank you for your responses and support!

---

### Author Rebuttal · Authors · 2023-08-10

We would like to express our sincere gratitude for your overall positive feedback and recognizing the significance of our contributions. We introduce a novel combinatorial bandit model, *the cascading contextual assortment bandit*, which generalizes two of the prominent existing combinatorial bandits, cascading and assortment bandits and also generalizes single-action selection bandits. Not only do we generalize these bandit problem settings, but also we tackle longstanding open problem with suboptimal dependence on the cascade length. Hence, we take on a more general and more difficult problem and propose provably efficient algorithms and salient features and improved analysis. We strongly believe that the new model, the algorithms, the regret analysis, and the approximate optimization guarantees that we provide in this paper offer meaningful contributions to the community.

Incorporating the review of Reviewer nP45, we have also included an additional comparison to the CombCascade in [12] and intend to include more comparisons with additional methods in a revised version. For this experiment, we set (1) the total number of items $N=10$, the length of the cascade $K=2$, the size of the assortment $M=2$, the dimension (for the feature vector and parameter) $d=5$ (see Figure 6) and (2) $N=15, K=2, M=2, d=10$ (see Figure 7). You can see the results in the attached pdf file. Our proposed algorithm UCB-CCA and UCB-CCA+ perform better than C$^3$-UCB in [16] and CombCascade in [12].

---

### Author Response · Authors · 2023-08-21
**Thank you and Key Contributions**

Dear Reviewers (and AC)

We extend our heartfelt gratitude to all the reviewers for their constructive feedback and their overall positive evaluations. We express special appreciation for the insightful discussions that we've had during this review period. As we wrap up this discussion phase, we wish to underscore the crucial contributions that our work brings to the forefront.

## Key Contributions

1. **Novel Combinatorial Bandit Model**: Our work introduces a new combinatorial bandit model that generalizes both cascading bandits and assortment bandits. This novel model finds practical applications across various domains. Given the practical and wide applicability of the proposed model, we believe that there will be more research efforts studying this new model.

2. **Novel UCB Algorithm**: We propose a novel UCB-based bandit algorithm tailored for the newly proposed contextual assortment bandit problem. Furthermore, we establish regret bounds that eliminate the persistent suboptimal dependence on cascade length and worst-case examination probability – a long-standing and unnecessary drawback that existed before. Remarkably, our novel algorithmic design, including the swapping technique, as well as the new analysis helps achieve these improvements. Thus, our work tackles a more intricate and comprehensive problem while simultaneously delivering stronger results.

3. **Improved Algorithm**: Our contributions include an improved algorithm that not only reduces dependence on the problem-specific parameter $\kappa$ in the regret bound but also does so without exacerbating other dependencies.

4. **Rigorous Approximation Guarantee**: We provide a rigorous guarantee for approximate combinatorial optimization. Through our work, we demonstrate that a greedy algorithm for the cascading assortment optimization problem yields a 0.5 approximation of the optimal solution (detailed in Appendix D). To our knowledge, this result stands out as the first rigorous proof of approximation guarantee for the contextual cascading bandit problem. We believe that this particular outcome holds independent significance and interest.

By consolidating these contributions where we believe each contribution is unique and valuable, we hold the firm belief that our paper as a whole introduces distinctive and substantial advancements to the existing literature.

Thank you once again for your efforts.

---

### Decision · Program_Chairs · 2023-09-21

**Decision:**

Accept (poster)

**Comment:**

The paper is a study on the mixture model that combines cascading bandits and assortment bandits together in a contextual bandit environment. The problem is reasonably motivated. The authors provide algorithms and theoretical analysis of their regret bounds.  All reviewers are positive in general to the paper (with scores 7, 6, 6, 5, 5), and the issues raised by reviewers have mostly been addressed by the authors rebuttal. Reviewer anEU did point out a recent ICML'23 paper that partially overlaps with the current submission. That paper can be viewed as a concurrent paper with the current one, and as the authors explained, two studies use quite different approaches, so the overlapping result does not post a serious issue here.

Given the overall positive support to the paper by all five reviewers, I recommend acceptance to the paper. I encourage the authors to provide a comprehensive revision to the paper to address all the comments, especially a detailed comparison with the concurrent and partially overlapping paper pointed out by a reviewer.